# Symmetries of the black hole interior and singularity regularization

**Marc Geiller, Etera R. Livine and Francesco Sartini**

Univ Lyon, ENS de Lyon, Univ Claude Bernard Lyon 1,
CNRS, Laboratoire de Physique, UMR 5672, Lyon, France

## Abstract

We reveal an $\mathfrak{iso}(2,1)$ Poincaré algebra of conserved charges associated with the dynamics of the interior of black holes. The action of these Noether charges integrates to a symmetry of the gravitational system under the Poincaré group ISO(2, 1), which allows to describe the evolution of the geometry inside the black hole in terms of geodesics and horocycles of AdS$_2$. At the Lagrangian level, this symmetry corresponds to Möbius transformations of the proper time together with translations. Remarkably, this is a physical symmetry changing the state of the system, which also naturally forms a subgroup of the much larger BMS$_3$ = Diff($S^1$) $\ltimes$ Vect($S^1$) group, where $S^1$ is the compactified time axis. It is intriguing to discover this structure for the black hole interior, and this hints at a fundamental role of BMS symmetry for black hole physics. The existence of this symmetry provides a powerful criterion to discriminate between different regularization and quantization schemes. Following loop quantum cosmology, we identify a regularized set of variables and Hamiltonian for the black hole interior, which allows to resolve the singularity in a black-to-white hole transition while preserving the Poincaré symmetry on phase space. This unravels new aspects of symmetry for black holes, and opens the way towards a rigorous group quantization of the interior.

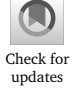

# 1 Introduction

Symmetries play a fundamental role in modern physics. They give rise to conservation laws via Noether's theorem [1], control the structure of classical solutions of a system, and also organize its quantum states via representation theory. They can be either as global or gauge symmetries, which can in turn be hidden, broken, restored, or deformed, and even have quantum realizations without classical analogue. General relativity is the typical example of a theory which rests solely on a principle of symmetry, namely that of invariance under space-time diffeomorphisms. This simplicity and elegance in the structure of the theory is however the tree hiding the forest, and it has been realized since the insight of Einstein that symmetries in general relativity play a very subtle role and are far from being completely understood.

Studying symmetries becomes particularly intriguing in the presence of boundaries. In gauge theories, boundaries can promote a subset of the gauge symmetries to global, physical symmetries, which in turn implies the existence of degenerate vacua [2–4]. For gravity in asymptotically flat spacetimes, the existence of these vacua is related to an enlargement of the symmetry group from Poincaré to the BMS group [5–7]. Such infinite-dimensional symmetry groups are in fact ubiquitous in gravity, and a whole zoology exists depending on the type and location of the boundary, the boundary conditions, the geometry and dimensionality of spacetime, and the formulation of gravity being considered. Interest in these symmetries lies in their (sometimes conjectural) ability to control gravitational scattering [2, 8, 9], to count states in black hole entropy [10–15], and to even define theories of quantum gravity [16–25].

Even setting aside the role of boundaries, bulk symmetries present their own subtleties. There exist many formulations of general relativity, which possess more or less gauge and in which diffeomorphism freedom may either be partially fixed or supplemented with additional gauge invariances [26–29], and it is not clear whether this freedom is actually innocent or not [30–32]. If, as made manifest in the presence of boundaries, gauge has indeed more

physical content than meets the eye [33–36], one should aim to understand in details the symmetry content of a given formulation.

In the context of symmetry-reduced models, it has recently been shown that homogeneous and isotropic cosmology coupled with a massless scalar field exhibits a 1-dimensional $SL(2,\mathbb{R})$ conformal invariance in the form of Möbius transformations of the proper time [37–41] (see also [42–46]). This symmetry also exists in Bianchi I models [38] and in the presence of a cosmological constant [41], and the relationship with this latter was in fact already pointed out by Gibbons in [47]. What comes as a surprise is that this conformal symmetry exists on top of the usual time reparametrization invariance, which itself is what survives of the spacetime diffeomorphisms in homogeneous cosmological models. In this sense this symmetry is "hidden". Furthermore, there is so far no systematic understanding of its origin and of whether it extends to other setups and embeds into a large symmetry. Perhaps not surprisingly however, in the analysis of this symmetry a central role is played by the Schwarzian derivative [48], whose relationship with conformal symmetry has been known ever since. This appearance of $SL(2,\mathbb{R})$ symmetry, of the underlying $AdS_2$ geometry, and of the Schwarzian, is very reminiscent of recent developments in 2-dimensional gravity and holography [49–56]. While this strongly suggests that there should be a relationship between these two bodies of work, the question remains open since the cosmological models studied in [37–41] have a very different geometrical setup from 2-dimensional gravity, and in particular do not a priori involve boundary physics.

In this article we study another homogeneous cosmological model[1], which is the black hole interior, or Kantowski–Sachs spacetime. Its 4-dimensional phase space is parametrized by two "positions" $(V_1, V_2)$ and their momenta $(P_1, P_2)$. We find that it naturally carries an $\mathfrak{iso}(2,1)$ algebra, which is that of the (2+1)-dimensional Poincaré group. As this algebra is given by the (semi-direct) sum $\mathfrak{sl}(2,\mathbb{R}) \oplus \mathbb{R}^3$, this provides in a sense an extension, including translations, of the results obtained for FLRW spacetimes in [37–41]. The $\mathfrak{sl}(2,\mathbb{R})$ piece is formed by what is now known as the CVH generators, namely the generator of phase space dilatations $C$, the position variable $V_2$ (which here for dimensional reasons we refrain from calling the volume), and the Hamiltonian $H$. The 3-dimensional Abelian part corresponding to the translations is generated by the position $V_1$, a constant of the motion, and $V_1 P_2$. At the level of the action, this results once again in an $SL(2,\mathbb{R})$ invariance under Möbius transformations of the proper time, and in addition to an invariance under translations of $V_2$. We compute the conserved Noether charges associated with these symmetries, and relate them to the phase space functions forming the above-mentioned $\mathfrak{iso}(2,1)$ algebra.

Interestingly, it is possible to show that the Poincaré transformations which leave the action invariant actually descend from the 3-dimensional BMS group [57–59]. This latter is given by the semi-direct product structure $BMS_3 = \text{Diff}(S^1) \ltimes \text{Vect}(S^1)$, whose factors act here respectively as arbitrary reparametrizations of the proper time and generalized translations (whose explicit form we give below). By studying the action of the BMS group by conjugation, we are led to interpret the variables $(V_1, V_2)$ as vector fields on the circle $S^1$ (which here is the compactified time axis), i.e. as elements of the $\mathfrak{bms}_3$ Lie algebra. While the theory itself is not invariant under BMS transformations, but only under Möbius transformations of proper time and some specific translations, this still suggests a possible important role of BMS symmetry in the black hole interior spacetime. There could for example exist a BMS-invariant formulation of the dynamics, or a boundary symmetry living e.g. on the horizon. As already mentioned above, the role of BMS (or Virasoro, or extensions thereof) symmetry in black hole spacetimes has been recognized long ago and studied in depth [15, 59–62], and recent work has also focused on cosmology [63, 64]. It is however at present not clear how to connect these results with our work. Indeed, what is surprising in our findings is *i*) that it is the 3-dimensional BMS

---

[1]By "cosmological" we mean symmetry-reduced, with finitely-many degrees of freedom.

group which seems to play a role even though we are in a 4-dimensional context, and $ii$) that we are not explicitly studying boundaries or boundary conditions. The answer to puzzle $i$) is probably that, due to the fact that we consider a homogeneous and spherically-symmetric model for the black hole interior, there could be a dimensional reduction at play and unsuspected connections with two or 3-dimensional gravity. Concerning point $ii$), we will see that there is a subtle sense in which a boundary actually *has* to be considered. This is because non-closed homogeneous cosmological models, like FLRW or the black hole interior which we study, require an IR cutoff in order for the spatial integrals to be well-defined and to obtain for the symmetry-reduced action (and symplectic structure and Hamiltonian) a mechanical model depending only on time. This cutoff, which we will introduce as a fiducial length $L_0$ restricting the radial integrals, plays a subtle role and appears in particular as a shift in the Hamiltonian constraint. It could be that homogeneous cosmological models are too "simple" in the sense that they blur the difference between bulk and boundary, which makes the contact with the topic of boundary symmetries evidently subtle. We will come back to this issue in future work, and for now continue to spell out the results of the present work.

Just like in the case of the homogeneous and isotropic FLRW model, the presence of an $\mathfrak{sl}(2, \mathbb{R})$ structure enables to reformulate the phase space dynamics of the black hole interior in terms of the exponentiated SL$(2, \mathbb{R})$ flow on the AdS$_2$ hyperboloid. This leads to an elegant geometrization of the dynamics in terms of horocycles (these are curves whose normal geodesics converge all asymptotically in the same direction). In addition, having the full Poincaré structure available on phase space open up the possibility of studying the quantization of the black hole interior in terms of representations of the algebra.

We finish this work by a discussion of the regularization of the black hole singularity. This is motivated by theories of quantum gravity such as loop quantum gravity (LQG) [65, 66], and its application to homogeneous symmetry-reduced models known as loop quantum cosmology (LQC) [67, 68]. In LQC, the Hamiltonian constraint is regularized by replacing a well-motivated choice of phase space variables, say $q$, by a compact periodic function such as $\sin(\lambda q)/\lambda$, where $\lambda$ is a (possibly phase space dependent) UV cutoff related to the minimal area gap of LQG. This procedure, known as "polymerization", emulates the fact that in full LQG the connection variable is defined in the quantum theory only through its holonomies (i.e. exponentiated operators). In the quantum theory, this leads to a quantization on a Hilbert space which is unitarily inequivalent to the Schrödinger representation used in Wheeler–DeWitt quantum cosmology, and in turn to a robust resolution of the big-bang singularity in FLRW models [69–71]. However, in the case of the black hole interior the choice of regularization scheme for the Hamiltonian constraint (i.e. the choice of phase space variables to polymerize and the expressions for the regulators) suffers from ambiguities which have so far not been settled [72–91]. While most schemes predict that the singularity is replaced by a bounce from the black hole to a white hole (see also [92–94] for a bounce towards a dS universe), there are disagreements as to which regularization to adopt, and as to the nature of the resulting effective spacetime geometry [95–102]. Luckily, symmetries provide a powerful criterion for reducing regularization and quantization ambiguities. Following [103], we find an LQC-like polymerization which preserves the Poincaré algebra structure on phase space. This is possible because polymerization is then simply viewed as a canonical transformation. This example of a polymerization scheme, which we study in section 6, is therefore in this sense compatible with the symmetries of the classical phase space. There exist however many other polymerization schemes (see references above), and one can ask if they are compatible with the Poincaré symmetry as well. This therefore provides a criterion to discriminate between the different schemes. We keep this investigation for future work. Sticking to the choice compatible with the Poincaré symmetry, the effective evolution shows that the polymerized model transitions via a bounce to a white hole. This therefore provides a black-to-white hole evolution which

preserves the symmetry on phase space, and opens up new interesting questions about the quantization of this model and its possible generalization.

The article is organized as follows. We start in section 2 by reviewing the classical setup for the study of the black hole interior dynamics as described by a Kantowski–Sachs spacetime. After solving the dynamics in the Hamiltonian formulation, we exhibit the $\mathfrak{iso}(2,1)$ algebra which controls this dynamics and the scaling properties of the phase space variables. In section 3, we then establish the relationship between this algebraic structure and classical symmetries of the action. In particular, we identify the Noether charges as the initial conditions for the $\mathfrak{iso}(2,1)$ generators. Section 4 is then devoted to the study of the BMS$_3$ group and of the embedding of the newly discovered Poincré symmetries of the action into it. In section 5 we endow (part of) the phase space with a geometrical structure, and reformulate the physical trajectories as curves on the hyperbolic AdS$_2$ plane. Finally, section 6 presents the construction of a symmetry-preserving regularization of the phase space, and studies the resulting black-to-white hole bouncing evolution. In appendix A we summarize some of the notations which are used throughout the paper.

## 2 Classical theory

We start by reviewing the Lagrangian and Hamiltonian formulations of the interior of Schwarzschild black holes. This enables us to introduce the phase space variables of the theory, along with the IR regulator needed in order to define it. We then solve the dynamics in the Hamiltonian picture, and show that this dynamics and the scaling properties of the phase space can be encoded in an $\mathfrak{iso}(2,1)$ Lie algebra. In the following section we then compute the Noether charges associated with the Poincaré symmetry of the action, and show that the $\mathfrak{iso}(2,1)$ algebra can equivalently be seen as generated by (evolving) constants of the motion.

In the region inside the horizon of a Schwarzschild BH, the radial direction becomes time-like, and on hypersurfaces orthogonal to this direction the metric is homogeneous. The whole interior can then be described by a Kantowski–Sachs cosmological spacetime with line element

$$ds^2 = -\left(\frac{2M}{T} - 1\right)^{-1} dT^2 + \left(\frac{2M}{T} - 1\right) dr^2 + T^2 d\Omega^2,\tag{2.1}$$

where $M$ is the BH mass and $d\Omega^2$ the metric on the 2-spheres at constant $r$ and $T$. In these homogeneous coordinates the spatial slices are $\Sigma = \mathbb{R} \times S^2$ with $r \in \mathbb{R}$, the singularity is located at $T = 0$, and the horizon is at $T = 2M$.

The metric (2.1) solves Einstein's equations, and in particular has a vanishing 4-dimensional Ricci scalar. In order to study the symmetries of the gravitational dynamics in the BH interior, as well as potential (quantum) corrections and regularizations, we want to construct a phase space. For this, we consider line elements of the more general form

$$ds^2 = -N^2 dt^2 + \frac{8V_2}{V_1} dx^2 + V_1 d\Omega^2,\tag{2.2}$$

where the metric components $N(t)$, $V_1(t)$, and $V_2(t)$ are three time-dependent functions. The lapse is dimensionless, while the $V_i$'s have the physical dimensions of areas.

Since the spatial geometry is homogeneous, integrals over non-compact spatial slices $\Sigma$ diverge. In order to compute the Einstein–Hilbert action, we therefore need to introduce a fiducial length $L_0$ which acts as an IR cutoff for the spatial integrations. Restricting the integration range to $x \in [0, L_0]$, the vacuum Einstein–Hilbert action integrated over $\mathcal{M} = \mathbb{R} \times [0, L_0] \times S^2$

and evaluated on (2.2) becomes

$$S = \frac{1}{16\pi} \int_{\mathcal{M}} \mathrm{d}^4 x \sqrt{-g}\, R = \int_{\mathbb{R}} \mathrm{d}t\, L_0 \left[ \frac{V_2(4N^2 V_1 + V_1'^2) - 2V_1 V_1' V_2'}{2\sqrt{2}N(V_1^3 V_2)^{1/2}} + \frac{\mathrm{d}}{\mathrm{d}t}\left( \frac{1}{\sqrt{2}N}\left(\sqrt{V_1 V_2}\right)' \right) \right],$$

(2.3)

where the prime $'$ denotes derivation with respect to $t$. We recognize in this action the two terms of the 3+1 ADM decomposition, namely the kinetic term in $1/N$ involving the time derivative of the 3-metric coming from the spatial curvature, and the potential term in $N$ without time derivatives. The boundary term appearing as a total time derivative in this action is precisely the Gibbons–Hawking–York term associated with the constant $t$ hypersurfaces $\Sigma$. We drop it in what follows since it does not play any role.

## 2.1 Hamiltonian formalism and classical solutions

With the action at our disposal, the next step is to compute the canonical momenta conjugated to the classical fields $V_i$, and to find the Hamiltonian. Dropping the boundary term, the Legendre transform is

$$S = \int \mathrm{d}t \left( P_i V_i' - \mathcal{H} \right),$$

(2.4)

where the canonical momenta are

$$P_1 = -\frac{L_0}{\sqrt{2}N\left(V_1^2 V_2\right)^{1/2}}\left(V_1 V_2' - V_1' V_2\right), \qquad P_2 = -\frac{L_0}{\sqrt{2}N\left(V_1^2 V_2\right)^{1/2}} V_1' V_1.$$

(2.5)

As usual, we do not assign a momentum to $N$ and treat it as a Lagrange multiplier. The Hamiltonian is

$$\mathcal{H} = -\frac{N}{L_0}\sqrt{\frac{2V_2}{V_1}}\left( L_0^2 + V_1 P_1 P_2 + \frac{1}{2}V_2 P_2^2 \right).$$

(2.6)

The classical theory is now described by a 4-dimensional phase space equipped with the Poisson brackets $\{V_i, P_j\} = \delta_{ij}$. As usual, the lapse enforces the scalar constraint $\mathcal{H} \approx 0$, and the time evolution of a phase space function $\mathcal{O}(t)$ is given by the Poisson bracket $\mathcal{O}' = \{\mathcal{O}, \mathcal{H}\}$.

It is now convenient to absorb the pre-factor in the expression (2.6) for the Hamiltonian by performing a redefinition of the lapse function such that

$$N = L_0 \sqrt{\frac{V_1}{2V_2}} N_{\mathrm{p}}, \qquad \mathcal{H} = N_{\mathrm{p}} H_{\mathrm{p}}.$$

(2.7)

This simplification of the Hamiltonian will turn out to be very useful for the analysis of the symmetries. In particular, note that it implies that the $L_0$ contribution to the Hamiltonian (2.6) becomes constant in phase space with the choice of lapse (2.7). This is the condition which will later on enable us to define a closed algebra of observables on phase space. Let us nevertheless emphasize that, although one does often perform such reparametrizations and changes of the lapse, this is actually a field-dependent redefinition[2], and is therefore only possible because

---

[2]This change of lapse function corresponds to a field-dependent time reparametrization

$$t \mapsto \bar{t}, \qquad \frac{\mathrm{d}t}{\mathrm{d}\bar{t}} = L_0 \sqrt{\frac{V_1}{2V_2}}\frac{N_{\mathrm{p}}}{N}.$$

Since the fields $V_1$ and $V_2$ are dynamical, this begs the question of whether or not the physics using $t$ or $\bar{t}$ are equivalent, especially at the quantum level where those fields acquire quantum fluctuations and it seems likely that a given time $t$ would correspond to a non-trivial probability distribution for $\bar{t}$ and vice-versa.

we do not impose specific boundary conditions on the lapse. Here we put aside the questions about such boundary conditions and the choice of physical time for cosmology, and consider this change of lapse as a mere technicality.

In terms of this redefined lapse function, the initial line element (2.2) becomes

$$ds^2 = -\frac{V_1 L_0^2}{2V_2} N_{\rm p}^2 dt^2 + \frac{8V_2}{V_1} dx^2 + V_1 d\Omega^2 \, , \tag{2.8}$$

leading to the reduced action (dropping once again the boundary term)

$$\mathcal{S} = \int dt \left[ N_{\rm p} L_0^2 + \frac{V_1'(V_2 V_1' - 2V_1 V_2')}{2N_{\rm p} V_1^2} \right] , \tag{2.9}$$

and the canonical momenta

$$P_1 = \frac{1}{N_{\rm p}} \frac{V_2 V_1' - V_1 V_2'}{V_1^2} , \qquad P_2 = -\frac{1}{N_{\rm p}} \frac{V_1'}{V_1} . \tag{2.10}$$

The lapse $N_{\rm p}$ is now a Lagrange multiplier imposing the Hamiltonian constraint $H_{\rm p} \approx 0$. As usual, this constraint generates the invariance of the action under time reparametrization

$$
\begin{aligned}
t & \mapsto \tilde{t} = f(t), \\
N_{\rm p} & \mapsto \widetilde{N}_{\rm p}(\tilde{t}) = f'(t)^{-1} N_{\rm p}(t), \\
V_i & \mapsto \widetilde{V}_i(\tilde{t}) = V_i(t).
\end{aligned}
\tag{2.11}
$$

This invariance means that we can completely reabsorb the lapse in a redefinition of the time coordinate. For this, we introduce the *proper time* (or comoving or cosmic time) $d\tau = N_{\rm p} dt$, and use the dot $\dot{\mathcal{O}} := d_\tau \mathcal{O}$ to denote derivation with respect to proper time. For an arbitrary phase space function $\mathcal{O}$, this gives the proper time evolution $\dot{\mathcal{O}} = \{\mathcal{O}, H_{\rm p}\}$ generated by the Hamiltonian constraint $H_{\rm p}$. With this redefinition of the time coordinate, the action now becomes

$$\mathcal{S} = \int d\tau \left[ L_0^2 + \frac{\dot{V}_1(V_2 \dot{V}_1 - 2V_1 \dot{V}_2)}{2V_1^2} \right] , \tag{2.12}$$

and the lapse $N_{\rm p}$ has therefore disappeared. Let us however emphasize that the lapse remains implicit in the definition of the integration variable $\tau$. It still plays the role of enforcing the Hamiltonian constraint $H_{\rm p} \approx 0$. If we were to truly forget about the lapse, $H_{\rm p}$ could take any arbitrary constant value, but here we still impose that $H_{\rm p}$ vanishes on-shell along the physical trajectories[3].

The Poisson brackets of the phase space variables with the Hamiltonian $H_{\rm p}$ now give the evolution equations

$$\dot{V}_1 = -V_1 P_2 \, , \tag{2.13a}$$

$$\dot{P}_1 = P_1 P_2 \, , \tag{2.13b}$$

$$\dot{V}_2 = -V_1 P_1 - V_2 P_2 \, , \tag{2.13c}$$

$$\dot{P}_2 = \frac{P_2^2}{2} \, . \tag{2.13d}$$

---

[3]Let us point out that, in the 4-dimensional scalar curvature, the term $L_0^2$ is the potential term coming from the 3-dimensional spatial curvature. Writing the action as an integral over proper time makes this term appear as a constant which therefore seems not to affect the equations of motion and the physical evolution. However, since a dynamical field (namely the lapse) is hidden in the definition of proper time $\tau$, the term $L_0^2$ nevertheless appears as a shift in the Hamiltonian. One should therefore be very careful with such constant/boundary terms, and how their status changes with changes of the time coordinate.

A straightforward calculation also reveals that the following quantities are first integrals of motion:

$$A = \frac{V_1 P_2^2}{2}, \qquad B = V_1 P_1, \qquad \{A, H_\mathrm{p}\} = 0 = \{B, H_\mathrm{p}\}. \qquad (2.14)$$

The equation of motion for $P_2$ can be directly integrated on its own. Then we use the two constants of the motion to obtain the evolution of $V_1$ and $P_1$. Finally, we use the fact that $H_\mathrm{p} = 0$ *on-shell* to find the trajectory for $V_2$. At the end of the day, we find that the evolution in proper time is given by

$$V_1 = \frac{A}{2}(\tau - \tau_0)^2, \qquad (2.15a)$$

$$P_1 = \frac{2B}{A(\tau - \tau_0)^2}, \qquad (2.15b)$$

$$V_2 = B(\tau - \tau_0) - \frac{L_0^2}{2}(\tau - \tau_0)^2, \qquad (2.15c)$$

$$P_2 = -\frac{2}{\tau - \tau_0}. \qquad (2.15d)$$

Inserting these solutions in the metric, and changing the variables as[4]

$$\tau - \tau_0 = \sqrt{\frac{2}{A}}\, T, \qquad x = \frac{1}{2L_0}\sqrt{\frac{A}{2}}\, r, \qquad (2.16)$$

we recover the standard Schwarzschild BH interior metric (2.1) with mass

$$M = \frac{B\sqrt{A}}{\sqrt{2} L_0^2}. \qquad (2.17)$$

Let us emphasize that this change of coordinates $(\tau, x) \to (T, r)$ contains a factor $A$, which is treated as a constant of the motion and not as a field-dependent phase space function $A = V_1 P_2^2 / 2$.

The 4-metric is singular for both $V_1 = 0$ and $V_2 = 0$. During the evolution, $\tau = \tau_0$ corresponds to the true singularity at $T = 0$, where both metric components vanish $V_1 = 0 = V_2$, while at the horizon singularity $T = 2M$ only the component $V_2 = 0$ vanishes. We draw the classical trajectories for different values of the initial conditions on figure 1.

Let us reflect on the apparent dependency on $L_0$ of the mass and evolution. Since this quantity was introduced as an IR regulator for the homogeneous metric, it should not play a role in the physical predictions, and consistently this is indeed the case. To make this explicit, notice that under a rescaling of the fiducial scale $L_0 \to \alpha L_0$ the canonical momenta scale as $P_i \to \alpha P_i$. This implies that the first integrals are also dilated as $A \to \alpha^2 A$ and $B \to \alpha B$, so that at the end of the day the mass (2.17) is indeed invariant under rescalings of the fiducial cell.

Another subtlety concerns the role of the constants of the motion. Despite the fact that there are two independent constants of the motion in the Hamiltonian, namely $A$ and $B$, once we plug the solutions for the configuration variables back in the metric line element (2.2), it turns out that only the combination $B\sqrt{A}$ is a physical observable. It does indeed measure the BH mass $M$ and the physical metric only depends on that parameter. We can nevertheless provide a physical interpretation for the first integral $B$. For this, notice that the vector field $\xi = \partial_x$ is a Killing vector for the family of metrics (2.8). The invariance of the metric

---

[4]Without loss of generality, the integration constant $\tau_0$ could be set to zero as a simple consequence of the gauge freedom in shifting time by a constant.

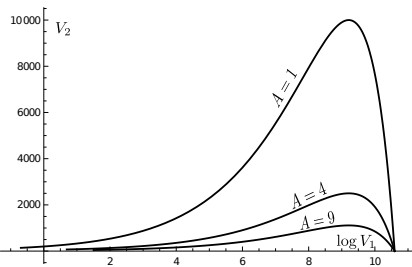

(a) Fixed mass $M = 100$ with varying $A = 1; 4; 9$.

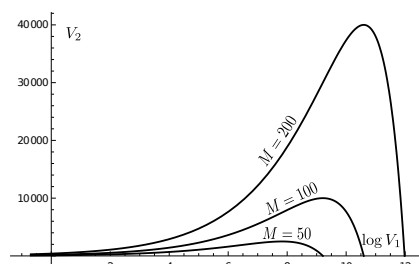

(b) Varying mass $M = 50; 100; 200$ with fixed $A = 1$.

Figure 1: Plot of the classical trajectories in configuration space $(\log V_1, V_2)$. The singularity is located at $V_1 = V_2 = 0$, while the BH horizon is at $V_2 = 0, V_1 \neq 0$. The fiducial length scale is set to $L_0 = 1$.

along this Killing vector gives a conserved Komar charge $Q_K$. To compute this charge, we introduce the determinant $q$ of the metric induced on the boundary 2-sphere, and the binormal $\epsilon^{\mu\nu} = (s^\mu n^\nu - s^\nu n^\mu)$ defined in terms of the time-like normal $n^\mu$ to the space-like foliation, and the space-like normal $s^\mu$ to the time-like boundary. The charge is then given by

$$Q_K = \frac{1}{8\pi} \int_{S^2} \mathrm{d}^2\Omega \sqrt{q}\, \epsilon^{\mu\nu} \nabla_\mu \xi_\nu = \frac{2}{L_0 N_p} \frac{V_2 V_1' - V_1 V_2'}{V_1} = 2\frac{V_1 P_1}{L_0} = 2\frac{B}{L_0}. \tag{2.18}$$

We indeed have that $\mathrm{d}_t Q_K = 0$ on-shell along physical trajectories. As for the mass, the charge $Q_K$ is invariant under rescalings of the fiducial length $L_0$. The difference with the BH mass is that $M$ is the Komar charge associated to the vector field $\partial_r$ in the Kantowski–Sachs metric (2.1), which differs from $\partial_x$ by the rescaling in (2.16). One can see that this rescaling changes $Q_K$ to the mass $M$.

## 2.2 The Poincaré algebra $\mathfrak{iso}(2, 1)$ encoding the dynamics

We now dig deeper into the algebraic structure of the evolution equations. The (proper) time variation of the metric components $V_1$ and $V_2$ is given by their Poisson brackets with the Hamiltonian density $H_p$. The second order time derivatives are then given by the iterated brackets $\{\{V_i, H_p\}, H_p\}$, and so on. One can then check if and at which stage this iteration closes and forms a Lie algebra. This gives the order of the differential equations which have to be solved in order to integrate the motion. In the present case, the generalized Kantowski–Sachs line element (2.2) is very similar to the homogenous metric of FLRW cosmology, and we might therefore expect a similar $\mathfrak{sl}(2, \mathbb{R})$ algebraic structure as that identified in [37–40]. We now show that this structure is actually extended to a Poincaré $\mathfrak{iso}(2, 1)$ Lie algebra encoding the dynamics of the BH interior.

Let us start with the metric component $V_2$. Its proper time variation is given by the Poisson bracket

$$\dot{V}_2 = \{V_2, H_p\} = C, \qquad C = -V_1 P_1 - V_2 P_2. \tag{2.19}$$

This phase space function $C$ is the generator of isotropic dilations of the phase space, as one can check that

$$e^{\{\eta C, \cdot\}} \triangleright P_i = e^{-\eta} P_i, \qquad e^{\{\eta C, \cdot\}} \triangleright V_i = e^\eta V_i, \qquad \forall i \in \{1, 2\}. \tag{2.20}$$

These transformations should not be confused with the physical rescalings of the fiducial length $L_0$. Indeed, here both the configuration variables and the momenta are rescaled by the finite

action of $C$, while if we change the size of the fiducial cell only the momenta are dilated. We come back to this subtle point in the next section. Now, let us consider the following modified Hamiltonian density obtained by a shift with the fiducial length:

$$H := H_{\mathrm{p}} + L_0^2 = -V_1 P_1 P_2 - \frac{V_2 P_2^2}{2}. \tag{2.21}$$

We then have that the three quantities $(C, V_2, H)$ form an $\mathfrak{sl}(2, \mathbb{R})$ Lie algebra, which we will refer to as the CVH algebra as in the previous work on FLRW cosmology [37–40]. The brackets are

$$\{C, V_2\} = V_2, \qquad \{V_2, H\} = C, \qquad \{C, H\} = -H. \tag{2.22}$$

The $\mathfrak{sl}(2, \mathbb{R})$ Casimir necessarily commutes with the Hamiltonian constraint, and one finds that it is related to the constant of the motion $B$ as

$$\mathcal{C}_{\mathfrak{sl}(2, \mathbb{R})} = -C^2 - 2HV_2 = -B^2 < 0. \tag{2.23}$$

An important difference with the previous work on FLRW cosmology nevertheless remains the shift by the fiducial length term $L_0^2$. In the cosmological context of an FLRW model, this term would actually correspond to a non-vanishing spatial curvature. This was not considered in [37–40] which focused on the spatially flat model. Here, going back to the initial unshifted Hamiltonian, it means that we are actually working with a centrally-extended $\mathfrak{sl}(2, \mathbb{R})$ Lie algebra with brackets

$$\{C, V_2\} = V_2, \qquad \{V_2, H_{\mathrm{p}}\} = C, \qquad \{C, H_{\mathrm{p}}\} = -H_{\mathrm{p}} - L_0^2. \tag{2.24}$$

When integrating the algebra into a group action, the extra central term simply produces phases which can easily be tracked.

Repeating the above construction starting from the other configuration variable $V_1$, we again obtain a closed Lie algebra, this time involving the other constant of the motion $A$ and a new phase space function $D$ defined as

$$V_1, \qquad A = \frac{V_1 P_2^2}{2}, \qquad D = V_1 P_2. \tag{2.25}$$

Putting this together, all the non-vanishing Poisson brackets of interest are

$$\begin{aligned}
\{C, V_2\} &= V_2, & \{C, V_1\} &= V_1, & \{D, H\} &= -A, \\
\{V_2, H\} &= C, & \{A, C\} &= A, & \{V_1, H\} &= -D, \\
\{C, H\} &= -H, & \{A, V_2\} &= -D, & \{D, V_2\} &= -V_1.
\end{aligned} \tag{2.26}$$

We recognize this as the Poincaré algebra $\mathfrak{iso}(2, 1)$ given by the semi-direct sum $\mathfrak{sl}(2, \mathbb{R}) \oplus \mathbb{R}^3$. Writing the $\mathfrak{sl}(2, \mathbb{R})$ commutators in the usual $\mathfrak{so}(2, 1)$ generator basis, one can write the brackets above in terms of the standard $\mathfrak{iso}(2, 1)$ generators

$$\begin{aligned}
j_z &= \frac{1}{\sqrt{2}}(V_2 - H), & k_x &= \frac{1}{\sqrt{2}}(V_2 + H), & k_y &= C, \\
\Pi_x &= D, & \Pi_y &= \frac{1}{\sqrt{2}}(V_1 - A), & \Pi_0 &= \frac{1}{\sqrt{2}}(V_1 + A),
\end{aligned} \tag{2.27}$$

for which we find

$$\begin{aligned}
\{j_z, k_i\} &= \epsilon_{ij} k_j, & \{k_x, k_y\} &= -j_z, \\
\{j_z, \Pi_i\} &= \epsilon_{ij} \Pi_j, & \{k_i, \Pi_0\} &= \Pi_i, & \{k_i, \Pi_j\} &= \delta_{ij} \Pi_0, \tag{2.28}
\end{aligned}$$

with $i, j \in \{x, y\}$. The two Poincaré Casimirs are given by

$$\mathcal{C}_1 = \Pi_0^2 - \Pi_x^2 - \Pi_y^2, \qquad \mathcal{C}_2 = j_z \Pi_0 + k_y \Pi_x - \Pi_y k_x. \qquad (2.29)$$

Expressing the generators in terms of phase space variables we find that both $\mathcal{C}_1$ and $\mathcal{C}_2$ are identically vanishing. These two Casimir conditions reduce the 6-dimensional Lie algebra $\mathfrak{iso}(2,1)$ back to the original 4-dimensional phase space generated by the configuration and momentum variables $(V_i, P_i)_{i=1,2}$. This also means that the mini-superspace black hole interior carries a massless unitary representation of the Lie algebra $\mathfrak{iso}(2,1)$. Preserving this structure at the quantum level means quantizing the black hole interior dynamics in terms of Poincaré representations, similarly to what has been developed for the FLRW cosmological model[5] in [38, 39].

The beautiful feature of this algebraic structure for the black hole dynamics in the Hamiltonian formalism is that the Poincaré algebra $\mathfrak{iso}(2,1)$ can be exponentiated into an actual symmetry of the Lagrangian. The Poincaré group symmetry $\mathrm{ISO}(2,1)$ is generated by the initial conditions for the evolution of the Poincaré generators in proper time, similarly to the construction done for the $\mathrm{SL}(2, \mathbb{R})$ symmetry in the cosmological case [39]. This is the subject of the next section.

As a remark, we note that here the Hamiltonian $H$ belongs to the non-Abelian $\mathfrak{sl}(2, \mathbb{R})$ subalgebra of the Poincaré algebra $\mathfrak{iso}(2,1)$. This is very different from the usual spacetime picture, where the Hamiltonian is part of the Abelian generators, and can be seen as a consequence of the fact that, as pointed out above, the Poincaré structure which we have discovered here has nothing to do per se with the isometries of spacetime. In the group quantization of the model using representation theory, this will have important consequences as it will determine how the Hamiltonian is represented.

# 3 Symmetries of the classical action

In this section, we show how the $\mathfrak{iso}(2,1)$ algebraic structure encoding the Hamiltonian dynamics generates an invariance of the action under $\mathrm{ISO}(2,1)$ Poincaré group transformations. On the one hand, the $\mathrm{SO}(2,1) \sim \mathrm{SL}(2,\mathbb{R})$ subgroup corresponds to Möbius transformations, for which the metric components transform as primary fields under conformal transformation of the proper time. On the other hand, the Abelian subgroup $\mathbb{R}^3$ defines a symmetry under special time-dependent field transformations. We compute the Noether charges corresponding to this symmetry under infinitesimal Poincaré transformations and show that we recover the $\mathfrak{iso}(2,1)$ generators (2.26) derived in the previous section.

## 3.1 Poincaré invariance: Möbius transformations and translations

### 3.1.1 The $\mathrm{SL}(2, \mathbb{R})$ invariance under Möbius transformations

Following the logic introduced for FLRW cosmology in [39, 41, 104], we start from the action (2.9) written as an integral over the proper time $\tau$, and show that it is invariant up to a

---

[5]In fact, one could ask whether it possible to identify an actual limit of the present construction which gives back FLRW cosmology. For instance, for an arbitrary integer $k \in \mathbb{N}$, one can take $V_1 = a^k L_0^2$, $k(6-k)V_2 = a^3 L_0^2$, $N_\mathrm{p} = N/L_0$ in (2.9) to obtain the action

$$S \to \int \mathrm{d}t \left[ L_0 N - V_0 \frac{3aa'^2}{8\pi N} \right],$$

where $V_0 = 4\pi L_0^3/3$ is the fiducial volume. Up to a constant shift in the energy, this is the reduced FLRW action. However, simply looking at the metric reveals that it is actually not possible to relate the two spacetimes, since e.g. in spherical coordinates the FLRW metric has $g_{\Omega\Omega} = r^2$ while in (2.8) this component is homogeneous.

boundary term under 1-dimensional conformal transformations. More precisely, we perform a Möbius transformation on the proper time $\tau$ and assume that the configuration fields $V_i$ transform as primary fields of weight 1. This transformation is

$$\tau \quad \mapsto \quad \tilde{\tau} = \frac{a\tau + b}{c\tau + d} \qquad \text{with} \qquad ad - bc = 1, \tag{3.1}$$
$$V_i \quad \mapsto \quad \widetilde{V}_i(\tilde{\tau}) = \frac{V_i(\tau)}{(c\tau + d)^2}.$$

The time derivatives transform as

$$\mathrm{d}\tau \quad \mapsto \quad \mathrm{d}\tilde{\tau} = \frac{\mathrm{d}\tau}{(c\tau + d)^2}, \tag{3.2}$$
$$\frac{\mathrm{d}\widetilde{V}_i}{\mathrm{d}\tilde{\tau}} \quad = \quad \frac{\mathrm{d}V_i}{\mathrm{d}\tau} - \frac{2c\,V_i}{c\tau + d},$$

while the action is invariant up to a total derivative and gives

$$\int \mathrm{d}\tilde{\tau}\,\mathcal{L}\big[\widetilde{V}_i(\tilde{\tau}), \dot{\widetilde{V}}_i(\tilde{\tau})\big] \quad = \quad \int \mathrm{d}\tilde{\tau}\left[L_0^2 + \frac{\dot{\widetilde{V}}_1(\widetilde{V}_2\dot{\widetilde{V}}_1 - 2\widetilde{V}_1\dot{\widetilde{V}}_2)}{2\widetilde{V}_1^2}\right] \tag{3.3}$$
$$= \quad \int \mathrm{d}\tau\left[\mathcal{L}\big[V_i(\tau), \dot{V}_i(\tau)\big] - \frac{\mathrm{d}}{\mathrm{d}\tau}\left(\frac{L_0^2}{c\tau + d} + \tau L_0^2 - \frac{2c\,V_2}{c\tau + d}\right)\right].$$

This total derivative defines a boundary term which enters the derivation of the Noether charges computed below.

Before moving on to the translation symmetry of the model, let us take a step back and reflect on the peculiar role of the fiducial length scale (or IR cutoff) $L_0$. The term $L_0^2$ in the Lagrangian is clearly not invariant under the Möbius transformations. It nevertheless does not spoil the theory's invariance under Möbius transformations since it only produces a function of $\tau$, which does not depend on the dynamical fields and can therefore be considered as a mere total derivative or boundary term. This observation does however open the door to three inter-related questions:

1. The Lagrangian term $L_0^2$ comes from the 3-dimensional spatial curvature, and is only a constant with respect to the time $\tau$ chosen as the integration variable. For instance, with respect to the original time coordinate $t$, this term involves the lapse $N$ as well as the metric components $V_1$ and $V_2$. What is so special about the choice of time $\tau$?

2. Since Möbius transformations map $L_0^2$ to a function of the proper time, it might be more natural to begin from the onset with a time-dependent cutoff $L_0(\tau)$. In that case, working with a time-dependent boundary defined by $L_0(\tau)$ would require to consider corner contributions to the action due to the non-orthogonality of the time-like boundary and the spatial slices (e.g. see the recent works on space-time corners and the Hayward term for general relativity [24, 105–107]).

3. Introducing an explicitly time-dependent fiducial length could mean deciding on some profile $L_0(\tau)$, inserting it in the symmetry-reduced action, and considering it as a forced system. A more natural option would be to consider $L_0(\tau)$ as a dynamical field. This amounts to considering the spatial boundary as a dynamical variable of the theory, with its own physical variation, equations of motion and quantum fluctuations. This is reminiscent of recent work on edge modes which suggest to consider the embedding variables defining the location of the boundary as fields [108–111].

We believe that this elucidation of the role of the IR cutoff will play an essential role in understanding the origin of the symmetries and possible holographic properties of the black hole interior dynamics. These aspects are however not essential for the rest of the present discussion, and we leave them for future investigation.

### 3.1.2 The $\mathbb{R}^3$ invariance under translations

On top of the invariance under conformal reparametrizations of the proper time, the classical action admits another symmetry. This symmetry does not modify the time coordinate nor the metric component $V_1$, but only $V_2$, and acts as

$$\tau \;\mapsto\; \tilde{\tau} = \tau\,, \tag{3.4a}$$
$$V_1 \;\mapsto\; \widetilde{V}_1(\tau) = V_1\,, \tag{3.4b}$$
$$V_2 \;\mapsto\; \widetilde{V}_2(\tau) = V_2 + (\alpha + \beta\tau + \gamma\tau^2)\dot{V}_1 - (\beta + 2\gamma\tau)V_1\,, \tag{3.4c}$$

where one should notice that $d_\tau(\alpha + \beta\tau + \gamma\tau^2) = (\beta + 2\gamma\tau)$. The induced variation of the action yields a total derivative as

$$\int d\tilde{\tau}\,\mathcal{L}\big[\widetilde{V}_i(\tilde{\tau}),\dot{\widetilde{V}}_i(\tilde{\tau})\big] = \int d\tau\left[\mathcal{L}\big[V_i(\tau),\dot{V}_i(\tau)\big] + \frac{d}{d\tau}\left(2\gamma V_1 - (\alpha+\beta\tau+\gamma\tau^2)\frac{\dot{V}_1^2}{2V_1}\right)\right], \tag{3.5}$$

meaning that these translation are indeed a symmetry of the theory. Below we compute the corresponding Noether charges and associated constants of the motion.

## 3.2 Infinitesimal transformations and Noether charges

According to Noether's theorem, if the action of a system with Lagrangian coordinates $V_i$ is invariant under the infinitesimal variations

$$\delta\tau = \tilde{\tau} - \tau\,, \qquad \delta V_i = \widetilde{V}_i(\tau) - V_i(\tau)\,, \tag{3.6}$$

in the sense that

$$\delta\mathcal{S} = \mathcal{S}\big[\tilde{\tau},\widetilde{V}_i(\tilde{\tau}),\dot{\widetilde{V}}_i(\tilde{\tau})\big] - \mathcal{S}\big[\tau,V_i(\tau),\dot{V}_i(\tau)\big] \approx \int d\tau\,\frac{dF}{d\tau}\,, \tag{3.7}$$

where the last relation holds *on-shell*, then the Noether charge defined as

$$Q := -\frac{\partial\mathcal{L}}{\partial\dot{V}_i}\delta V_i - \mathcal{L}\delta\tau + F \tag{3.8}$$

is a constant of the motion along classical trajectories.

Let us start by identifying the charges corresponding to the conformal symmetry. An arbitrary Möbius transformation can be decomposed in terms of three types of transformations: translations, dilations and special conformal transformations. Their infinitesimal generators are given for an infinitesimal parameter $\varepsilon \to 0$ by the following choice of Möbius parameters:

- — Translations:        $a = 1,\quad b = \varepsilon,\quad c = 0,\quad d = 1,$
- — Dilations:             $a = 1/d = \sqrt{1+\varepsilon},\quad b = 0,\quad c = 0,$
- — Special conformal: $a = 1,\quad b = 0,\quad c = \varepsilon,\quad d = 1.$

A straightforward calculation gives the infinitesimal variations of the proper time and of the metic components under these transformations, as well as the variation of the action given by

the total derivative term $F$. We find

$$
\begin{aligned}
\delta_\varepsilon^{(\mathrm{t})}\tau &= \varepsilon\,, & \delta_\varepsilon^{(\mathrm{t})}V_i &= -\varepsilon\dot{V}_i\,, & F^{(\mathrm{t})} &= \varepsilon\,, \\
\delta_\varepsilon^{(\mathrm{d})}\tau &= \varepsilon\tau\,, & \delta_\varepsilon^{(\mathrm{d})}V_i &= \varepsilon(V_i - \tau\dot{V}_i)\,, & F^{(\mathrm{d})} &= \varepsilon\tau\,, \\
\delta_\varepsilon^{(\mathrm{s})}\tau &= -\varepsilon\tau^2\,, & \delta_\varepsilon^{(\mathrm{s})}V_i &= -\varepsilon\tau(2V_i - \tau\dot{V}_i)\,, & F^{(\mathrm{s})} &= -\varepsilon\tau^2 + 2\varepsilon V_2\,.
\end{aligned}
\tag{3.9}
$$

We then find that the associated conserved charges are given by

$$
\begin{aligned}
Q_- &= \frac{\dot{V}_1 V_2 - 2V_1\dot{V}_2}{2V_1^2}\,, \\
Q_0 &= \tau\left(\frac{\dot{V}_1 V_2 - 2V_1\dot{V}_2}{2V_1^2}\right) + \dot{V}_2 = \dot{V}_2 + \tau Q_-\,, \\
Q_+ &= -\tau^2\left(\frac{\dot{V}_1 V_2 - 2V_1\dot{V}_2}{2V_1^2}\right) - 2\tau\dot{V}_2 + 2V_2 = 2V_2 - 2\tau Q_0 + \tau^2 Q_-\,,
\end{aligned}
\tag{3.10}
$$

which generate respectively the translations, dilations, and special conformal transformations. The conservation of these charges can be verified explicitly using the solutions (2.15) to the equations of motion found in the previous section. This gives the *on-shell* values of the charges

$$
Q_- \approx L_0^2\,, \qquad Q_0 \approx B\,, \qquad Q_+ \approx 0\,.
\tag{3.11}
$$

The last step is now to translate these charges into the Hamiltonian formalism, and to express them in terms of the canonical variables. Translating the time derivatives $\dot{V}_i$ into the conjugate momenta $P_i$, we find that the Noether charges admit a remarkably simple expression in terms of the CVH generators, given by

$$
Q_- = H\,, \qquad Q_0 = C + \tau H\,, \qquad Q_+ = 2V_2 - 2\tau C - \tau^2 H\,.
\tag{3.12}
$$

Keeping in mind the fact that the time evolution of a phase space function $\mathcal{O}$ is given by the expression $\dot{\mathcal{O}} = \{\mathcal{O}, H_\mathrm{p}\} + \partial_\tau\mathcal{O} = \{\mathcal{O}, H\} + \partial_\tau\mathcal{O}$, it is straightforward to check directly the conservation of these charges. Since they explicitly depend on $\tau$, these are actually evolving constants of the motion.

Finally, we remark that the Noether charges are indeed the generators of the infinitesimal Möbius transformation. Computing their bracket with the fields $V_i$, and replacing the momenta with their expression in terms of derivative of $V_i$, gives back the variations (3.9). In fact, the Noether charges are actually the initial conditions for the three observables $H$, $C$ and $V_2$. Indeed, if we reverse the expression of the charges given above, we obtain the trajectories for the CVH observables given by

$$
H = Q_-\,, \qquad C = Q_0 - \tau Q_-\,, \qquad V_2 = Q_+ + \tau Q_0 - \frac{1}{2}\tau^2 Q_-\,.
\tag{3.13}
$$

Let us now look into the Abelian symmetry (3.4). We distinguish the three types of transformations in terms of the degree of the polynomial in $\tau$. Their infinitesimal versions and the resulting variation of the Lagrangian given by the total derivative F are

$$
\begin{aligned}
\delta_\varepsilon^{(-)}V_2 &= \varepsilon\dot{V}_1\,, & F^{(-)} &= -\varepsilon\frac{\dot{V}_1^2}{2V_1}\,, \\[2mm]
\delta_\varepsilon^{(0)}V_2 &= -\varepsilon(V_1 - \tau\dot{V}_1)\,, & F^{(0)} &= -\varepsilon\tau\frac{\dot{V}_1^2}{2V_1}\,, \\[2mm]
\delta_\varepsilon^{(+)}V_2 &= \varepsilon\tau(2V_1 - \tau\dot{V}_1)\,, & F^{(-)} &= -2\varepsilon V_1 + \varepsilon\tau^2\frac{\dot{V}_1^2}{2V_1}\,,
\end{aligned}
\tag{3.14}
$$

where $(-, 0, +)$ represent respectively the transformations of degree $(0, 1, 2)$. We recall that for the translations there is no variation of the proper time $\tau$ or of the metric component $V_1$. We now compute the corresponding Noether charges and express them as functions on the phase space to find

$$P_- = A, \qquad P_0 = D + \tau A, \qquad P_+ = -2V_1 - 2\tau D - \tau^2 A, \qquad (3.15)$$

which means that the Noether charges $P$ are the initial conditions for the three observables $A$, $D$ and $V_1$. Moreover, computing their on-shell value gives

$$P_- \approx A, \qquad P_0 \approx 0, \qquad P_+ \approx 0. \qquad (3.16)$$

Finally, it is straightforward to check that the Poisson brackets with the $P$'s do generate the infinitesimal variations of the field translations (3.14).

With the expressions (3.12) and (3.15) for the charges in terms of the canonical variables, it is direct to verify that the Noether charges reproduce the $\mathfrak{iso}(2, 1)$ algebra

$$
\begin{aligned}
&\{Q_0, Q_\pm\} = \pm Q_\pm, & &\{Q_+, Q_-\} = 2Q_0, \\
&\{Q_0, P_\pm\} = \pm P_\pm, & &\{Q_0, P_0\} = 0, \\
&\{Q_\pm, P_\pm\} = 0, & &\{Q_\pm, P_\mp\} = \pm 2P_0, \\
&\{Q_\pm, P_0\} = \mp P_\pm, & &\{P_I, P_J\} = 0, \qquad I, J = 0, \pm.
\end{aligned}
\qquad (3.17)
$$

We have shown that the $Q$'s and $P$'s are the initial conditions for the phase space observables $\{V_1, V_2, H, C, A, D\}$. Conversely, the observables $\{V_1, V_2, H, C, A, D\}$ are the evolving version of the Noether charges, i.e. they are obtained by the exponentiated action of the constraint operator $\exp\big(\{\tau H, \bullet\}\big)$ on the $Q$'s and $P$'s. This explain why the observables $\{V_1, V_2, H, C, A, D\}$ form a closed $\mathfrak{iso}(2, 1)$ Lie algebra, which is to be understood as the dynamical version of the Poincaré algebra for the conserved charges derived above.

This shows how the $\mathfrak{iso}(2, 1)$ Lie algebra of observables encodes both the dynamics and the scaling properties of the phase space and comes from the invariance of the Kantowski–Sachs action under conformal transformations in proper time and under the field translations (3.4). We will furthermore show in section 4 that these symmetry transformations form a subgroup of the more general BMS transformations acting as proper time reparametrizations.

## 3.3 Action on the physical trajectories

Before moving on to closer mathematical investigation of the group properties of the symmetry transformations, we would like to understand how these transformations act on the physical trajectories of the system. In particular, we want to determine if they simply consist in a different time parametrization of the same trajectories, or if they map between different trajectories. Since different trajectories are labelled by the values of the first integrals $A$ and $B$, we will study how these are modified by the conformal transformations in proper time and field translations.

Möbius reparametrizations of the proper time (3.1) are symmetries, which means that they map classical solutions onto classical solutions. Computing their explicit action on a physical

trajectory (2.15) gives

$$V_1(\tau) = \frac{A}{2}\tau^2 \quad \mapsto \quad \widetilde{V}_1(\tilde{\tau}) = \frac{A}{2}(d\tilde{\tau} - b)^2, \tag{3.18}$$

$$V_2(\tau) = \tau\left(B - L_0^2 \frac{\tau}{2}\right) \quad \mapsto \quad \widetilde{V}_2(\tilde{\tau}) = (d\tilde{\tau} - b)\left(B(a - c\tilde{\tau}) - \frac{L_0^2}{2}(d\tilde{\tau} - b)\right), \tag{3.19}$$

$$P_1(\tau) \quad \mapsto \quad \widetilde{P}_1(\tilde{\tau}) = \frac{2B}{A}\frac{1}{(d\tilde{\tau} - b)^2}, \tag{3.20}$$

$$P_2(\tau) \quad \mapsto \quad \widetilde{P}_2(\tilde{\tau}) = \frac{2d}{b - d\tilde{\tau}}, \tag{3.21}$$

where we have used the inverse Möbius transformation to (3.1), given by $\tau = (d\tilde{\tau} - b)/(a - c\tilde{\tau})$. The new metric components $\widetilde{V}_i(\tilde{\tau})$ are obviously still solutions to the equations of motion, but the values of the constants of the motion need to be slightly adjusted. Actually, closer inspection reveals that the value of $B$ remains the same while the value of $A$ acquires a transformation-dependent factor. Explicitly, we have

$$A = \frac{V_1 P_2^2}{2} \mapsto \tilde{A} = \frac{\widetilde{V}_1 \widetilde{P}_2^2}{2} = d^2 A, \qquad\qquad B = V_1 P_1 \mapsto \widetilde{B} = \widetilde{V}_1 \widetilde{P}_1 = B. \tag{3.22}$$

This result is consistent with the fact that $B$ is the Casimir invariant of the $\mathfrak{sl}(2,\mathbb{R})$ algebra, and as such is expected to be conserved under $\mathrm{SL}(2,\mathbb{R})$ transformations. On the other hand, $A$ belongs to the translational sector of $\mathfrak{iso}(2,1)$ and is therefore naturally modified by this symmetry.

However, the story of the conformal mapping is actually more subtle because it happens to shift the Hamiltonian constraint, which therefore does not seem to vanish anymore. Indeed, we have

$$H_{\mathrm{p}} = -L_0^2 - \frac{P_2}{2}(2V_1 P_1 + V_2 P_2) = 0 \quad \mapsto \quad \widetilde{H}_{\mathrm{p}} = -L_0^2 - \frac{\widetilde{P}_2}{2}(2\widetilde{V}_1\widetilde{P}_1 + \widetilde{V}_2\widetilde{P}_2) = 2cdB + (d^2 - 1)L_0^2.$$

This apparent puzzle is resolved by the fact that the fiducial length does actually change under conformal transformations. More precisely, in order to interpret the new trajectory as a black hole solution, we need to restore the on-shell condition and redefine the fiducial scale as

$$L_0^2 \mapsto \widetilde{L}_0^2 = 2cdB + d^2 L_0^2. \tag{3.23}$$

Indeed, one can see that this requirement ensures that

$$H_{\mathrm{p}} = 0 \mapsto \widetilde{H}_{\mathrm{p}} = -\widetilde{L}_0^2 - \frac{\widetilde{P}_2}{2}(2\widetilde{V}_1\widetilde{P}_1 + \widetilde{V}_2\widetilde{P}_2) = 0. \tag{3.24}$$

Because of the presence of the fiducial length in the line element (2.8), the 4-metric gets modified as well, and becomes

$$d\tilde{s}^2 = -\frac{\widetilde{V}_1 \widetilde{L}_0^2}{2\widetilde{V}_2}d\tilde{\tau}^2 + \frac{8\widetilde{V}_2}{\widetilde{V}_1}dx^2 + \widetilde{V}_1 d\Omega^2. \tag{3.25}$$

Inserting the explicit trajectories in $\tilde{\tau}$ in this expression and performing the change of coordinates

$$t = \sqrt{\frac{A}{2}}(d\tilde{\tau} + b), \qquad\qquad x = \frac{d}{2\widetilde{L}_0}\sqrt{\frac{A}{2}}\,r, \tag{3.26}$$

gives the Kantowski–Sachs metric with a dilated mass

$$M_{\text{BH}} \mapsto \widetilde{M}_{\text{BH}} = \frac{dL_0^2}{\widetilde{L}_0^2} M_{\text{BH}} = \frac{dB\sqrt{A}}{\sqrt{2}\widetilde{L}_0^2}. \tag{3.27}$$

This shows that the Möbius symmetry transformations are not mere time reparametrizations of the classical solutions, but actually map physical black hole trajectories onto different trajectories with different initial conditions (the values of $A$ and $B$) and a different black hole mass.

We treat the $\mathbb{R}^3$ symmetry transformations in the same manner. Field translations (3.4) act only on $V_2$. The resulting flow between trajectories corresponds to a simple shift in the constants of the motion, given by

$$A \mapsto \widetilde{A} = A, \qquad B \mapsto \widetilde{B} = B + \alpha A. \tag{3.28}$$

In order to preserve the Hamiltonian constraint and stay on-shell we need to modify the fiducial length as

$$L_0^2 \mapsto \widetilde{L}_0^2 = L_0^2 - \beta A, \tag{3.29}$$

which in turn leads to a shifted BH mass

$$M_{\text{BH}} \mapsto \widetilde{M}_{\text{BH}} = \frac{1}{\widetilde{L}_0^2}\left( L_0^2 M_{\text{BH}} + \alpha \frac{A^{3/2}}{\sqrt{2}} \right). \tag{3.30}$$

This shows that the Poincaré symmetry transformations given by both Möbius transformations and field translations, are not simple trajectory reparametrizations but more generally allow to flow between black hole solutions with different masses and explore the *physical* phase space.

# 4 Poincaré group and BMS transformations

We have shown in the previous section that the Kantowski–Sachs mini-superspace for the black hole interior metric admits a conformal and translational invariance on top of the diffeomorphism symmetry of general relativity. Indeed, as we have just seen in section 3.3, the conformal and translational invariance act non-trivially on the physical parameter labelling the solutions (i.e. the mass), at the difference with diffeomorphisms. The present section is dedicated to the deeper analysis of the group properties of these new symmetry transformations. Indeed, we have introduced and studied in totally independent ways the Möbius SL(2,$\mathbb{R}$) transformations and the $\mathbb{R}^3$ field translations acting on the action, but also shown that their infinitesimal variations couple to each other to form an $\mathfrak{iso}(2,1)$ Lie algebra. Here, we explicitly show that the finite transformations form an ISO(2,1) Poincaré group. Moreover, this is realized by the natural embedding of this group into the larger group BMS$_3$ = Diff($S^1$) $\ltimes$ Vect($S^1$). It is compelling to consider this BMS group as the fundamental symmetry group of the black hole mini-superspace dynamics.

Note however, that the BMS group identified here has a priori nothing to do with the "usual" BMS group of spacetime symmetries which extends the Poincaré group of isometries. The BMS group structure does indeed appear here, but it is not inherited from e.g. the asymptotic symmetries of the spacetime. In particular, the translations in Vect($S^1$) are not a priori related to the translations of the *spacetime* BMS group, but instead simply correspond to the Abelian part of the symmetry which we unravel.

## 4.1 Extended Poincaré transformations and the BMS$_3$ group

As we have introduced Möbius reparametrizations of the proper time in (3.1), it is natural to extend these to *arbitrary* reparametrizations of the proper time. This follows the work done on the conformal invariance of FLRW cosmology [39, 41, 47, 104]. We therefore consider a general reparametrization of the proper time, and assume that the metric components $V_i$ are primary fields of weight 1. This is

$$
D_f : \quad \left| \begin{array}{rcl} \tau & \mapsto & \tilde{\tau} = f(\tau), \\ V_i & \mapsto & \widetilde{V}_i(\tilde{\tau}) = \dot{f}(\tau) V_i(\tau), \end{array} \right. \qquad \text{thus} \qquad \left| \begin{array}{rcl} \mathrm{d}\tau & \mapsto & \mathrm{d}\tilde{\tau} = \dot{f}\,\mathrm{d}\tau, \\ \mathrm{d}_\tau V_i & \mapsto & \mathrm{d}_{\tilde{\tau}}\widetilde{V}_i = \mathrm{d}_\tau V_i + V_i\,\mathrm{d}_\tau \ln \dot{f}. \end{array} \right.
\tag{4.1}
$$

The resulting variation of the action is a volume term weighted by a Schwarzian derivative, plus a boundary term given by a total derivative, i.e.

$$
\Delta_f \mathcal{S} = \mathcal{S}\big[\widetilde{V}_i(\tilde{\tau}), \dot{\widetilde{V}}_i(\tilde{\tau})\big] - \mathcal{S}\big[V_i(\tau), \dot{V}_i(\tau)\big] = \int \mathrm{d}\tau \left[ \mathrm{Sch}[f] V_2 + \frac{\mathrm{d}}{\mathrm{d}\tau}\left( L_0^2(f - \tau) - \frac{\ddot{f}}{\dot{f}} V_2 \right) \right],
\tag{4.2}
$$

where the Schwarzian derivative $\mathrm{Sch}[\cdot]$ is defined as

$$
\mathrm{Sch}[f] = \frac{f^{(3)}}{\dot{f}} - \frac{3}{2}\left(\frac{\ddot{f}}{\dot{f}}\right)^2 = \mathrm{d}_\tau^2 \ln \dot{f} - \frac{1}{2}\left(\mathrm{d}_\tau \ln \dot{f}\right)^2.
\tag{4.3}
$$

Such a transformation is therefore a classical symmetry only if the bulk variation vanishes, which requires the Schwarzian derivative $\mathrm{Sch}[f]$ to vanish. This happens if and only if $f$ is a Möbius transformation, i.e.

$$
\mathrm{Sch}[f] = 0 \qquad \Longleftrightarrow \qquad \exists(a, b, c, d),\ f(\tau) = \frac{a\tau + b}{c\tau + d}.
\tag{4.4}
$$

This shows that the Möbius transformations (3.1) introduced in the previous section are indeed the only case in which the proper time reparametrizations are a symmetry of the theory.

We can similarly extend the $\mathbb{R}^3$ field translations (3.4) and consider the more general transformations parametrized by an arbitrary function $g(\tau)$ and acting as

$$
T_g : \quad \left| \begin{array}{rcl} \tau & \mapsto & \tilde{\tau} = \tau, \\ V_1 & \mapsto & \widetilde{V}_1(\tau) = V_1, \\ V_2 & \mapsto & \widetilde{V}_2(\tau) = V_2 + g\dot{V}_1 - V_1\dot{g}. \end{array} \right.
\tag{4.5}
$$

Under this transformation, the induced variation of the action is a volume term weighted by the third derivative of the transformation parameter, plus a boundary term, i.e.

$$
\Delta_g \mathcal{S} = \mathcal{S}\big[\widetilde{V}_i(\tilde{\tau}), \dot{\widetilde{V}}_i(\tilde{\tau})\big] - \mathcal{S}\big[V_i(\tau), \dot{V}_i(\tau)\big] = \int \mathrm{d}\tau \left[ -g^{(3)} V_1 + \frac{\mathrm{d}}{\mathrm{d}\tau}\left( \ddot{g} V_1 - \frac{g\dot{V}_1^2}{2V_1} \right) \right].
\tag{4.6}
$$

Such a transformation is therefore a classical symmetry only if the third derivative vanishes, $g^{(3)} = 0$, which means that $g(\tau)$ is a second degree polynomial in the proper time, $g(\tau) = \gamma\tau^2 + \beta\tau + \alpha$. This gives back precisely the $\mathbb{R}^3$ field translations parametrized by three real numbers considered in (3.4).

A beautiful point is that these extended transformations, namely the conformal reparametrizations (4.1) and the field translations (4.5), actually form a group $\mathrm{Diff}(S^1) \ltimes \mathrm{Vect}(S^1)$, which one can recognize as the BMS$_3$ group [57–59]. As mentioned in the introduction, this group usually appears in the study of the symmetries of asymptotically flat

spacetimes [19, 20, 22, 112], where it produces an infinite-dimensional enhancement of the Poincaré group of isometries (like the two copies of Virasoro extend the two copies of $SL(2,\mathbb{R})$ in AdS$_3$ [16]). It is also heavily suspected to play a fundamental role in the presence of black holes [3, 15, 60, 113–117], in cosmological contexts [61, 63, 64], and has been shown to be related to fluid symmetries [118, 119]. Given this abundant literature, it is intriguing to find the BMS group in the new context described here. This happy coincidence points towards a deeper symmetry principle for the black hole mini-superspace. Furthermore, it turns out that the two volume terms appearing in the variation of the action above, namely the Schwarzian derivative Sch[$f$] and the third derivative $g^{(3)}$, are actually the group cocycle and the algebra cocycle corresponding to the central charges of the BMS$_3$ group. This definitely suggests that there should be a BMS group reformulation of the Kantowski–Sachs spacetime geometry. This line of research is postponed to future investigation.

For the time being, let us thus check the group structure of the extended transformations. To start with, on the one hand, the field translations form on their own an Abelian group with a simple composition law

$$T_{g_2} \circ T_{g_1} = T_{g_2 + g_1} \,. \tag{4.7}$$

This is the group Vect($S^1$) of 1-dimensional vector fields. On the other hand, the proper time reparametrizations also form a group with composition law

$$D_{f_2} \circ D_{f_1} = D_{f_2 \circ f_1} \,, \tag{4.8}$$

which we recognize as the group Diff($S^1$) of diffeomorphisms of the circle. The coupling between the two types of transformations comes from the non-trivial conjugation of a translation by a reparametrization, for which we obtain

$$D_{f^{-1}} \circ T_g \circ D_f = T_{(g \circ f)/\dot{f}} \,. \tag{4.9}$$

Putting conformal reparametrizations together with the field translations, we can consider pairs of transformations $(T_g, D_f) \in$ Vect($S^1$) $\times$ Diff($S^1$), which we define as the composed transformation

$$(T_g, D_f) := T_g \circ D_f \,. \tag{4.10}$$

The conjugation formula allows to compute the general composition laws for such pairs, which is given by

$$\begin{aligned} (T_{g_2} \circ D_{f_2}) \circ (T_{g_1} \circ D_{f_1}) &= T_{g_2} \circ (D_{f_2} \circ T_{g_1} \circ D_{f_2}^{-1}) \circ (D_{f_2} \circ D_{f_1}) \\ &= T_{g_2 + \dot{f}_2 \, g_1 \circ f_2^{-1}} \circ D_{f_2 \circ f_1} \,, \end{aligned} \tag{4.11}$$

along with the inverse transformation

$$(T_g \circ D_f)^{-1} = D_{f^{-1}} \circ T_{-g} = T_{-(g \circ f)/\dot{f}} \circ D_{f^{-1}} \,. \tag{4.12}$$

This is the group multiplication and its inverse for the Lie group defined as a semi-direct product Vect($S^1$) $\rtimes$ Diff($S^1$).

## 4.2 Adjoint BMS transformations

An enlightening point of view is to reverse the logic presented above. More precisely, we would like to start with the Diff($S^1$) $\ltimes$ Vect($S^1$) structure and the group multiplication (4.11), and then derive the field transformations (4.1) and (4.5) for the metric components. The

question is what are the metric components $V_1$ and $V_2$ with respect to the $\text{Diff}(S^1) \ltimes \text{Vect}(S^1)$ structure. The key insight is that the transformation law (4.1) of the metric components under a proper time reparametrization $D_f$ is exactly the transformation law for vector fields on the circle under a diffeomorphism in $\text{Diff}(S^1)$. Since both Lie groups $\text{Vect}(S^1)$ and $\text{Diff}(S^1)$ are generated by vector fields on $S^1$, it is tempting to imagine $V_1$ and $V_2$ as the $\mathfrak{bms}$ Lie algebra elements living in the adjoint representation with the BMS group elements $(T_g, D_f)$ acting on them by conjugation. This can be confirmed by a straightforward calculation, as we now demonstrate.

Let us consider the action by conjugation of an arbitrary BMS transformation $(T_g, D_f)$ on an infinitesimal BMS transformations $(T_{v_2}, D_{v_1})$,

$$(T_g, D_f) \rhd (T_{v_2}, D_{v_1}) = (T_g, D_f)(T_{v_2}, D_{v_1})(T_g, D_f)^{-1} \qquad \text{with} \quad \begin{vmatrix} v_1(\tau) &=& \tau + \varepsilon V_1(\tau) \\ v_2(\tau) &=& -\varepsilon V_2(\tau) \end{vmatrix},$$

(4.13)

for an infinitesimal parameter $\varepsilon \to 0$. Starting with the action of a circle diffeomorphism, we compute

$$D_f \circ (T_{v_2} \circ D_{v_1}) \circ D_{f^{-1}} = T_{\dot{f} \, v_2 \circ f^{-1}} \circ D_{f \circ v_1 \circ f^{-1}} = T_{v_2^{(f)}} \circ D_{v_1^{(f)}},$$

(4.14)

giving for the Lie algebra elements the variation

$$\forall \, i = 1, 2, \qquad V_i^{(f)}(\tau) = \dot{f}\big(f^{-1}(\tau)\big) V_i\big(f^{-1}(\tau)\big), \quad \text{or equivalently} \quad V_i^{(f)}\big(f(\tau)\big) = \dot{f} \, V_i(\tau).$$

(4.15)

This is exactly the transformation law (4.1). Similarly, we compute the action of a circle vector field

$$T_g \circ (T_{v_2} \circ D_{v_1}) \circ T_{-g} = T_{v_2 + g - \dot{v}_1 \, g \circ v_1^{-1}} \circ D_{v_1}.$$

(4.16)

The parameter $V_1$ clearly does not vary, while we calculate the transformation for the parameter $V_2$ to find

$$\varepsilon V_2^{(g)} = -v_2^{(g)} = -v_2 - g + \dot{v}_1 \, g \circ v_1^{-1} = \varepsilon(V_2 + g \dot{V}_1 - V_1 \dot{g}),$$

(4.17)

which reproduces the expected field translations (4.5).

This confirms the interpretation of the metric coefficients $V_1$ and $V_2$ as vector fields on the circle $S^1$. Let us remember that here this circle is simply the compactified time axis. This seems to imply a possible reformulation of the black hole reduced action in terms of differential forms on $S^1$, and one might wonder if this could then extend to full general relativity.

It is important to notice that here the $\text{BMS}_3$ group is acting on the metric coefficients $V_1$ and $V_2$ via the adjoint action. This is to be contrasted with the "usual" situation of geometric actions, where the asymptotic symmetry group (i.e. BMS or Virasoro) act by the *coadjoint action* instead [58, 120–122]. This enables a symplectic structure to be induced on the orbits, and therefore suggests that this might not be possible in the present situation. It is however possible to construct a geometric action where the conjugate variables $P_1$ and $P_2$ transform under the coadjoint action, although these are not the components which appear in the metric [123]. Alternatively, it could be interesting to study the "dual BMS group" $\text{Diff}(S^1) \ltimes \big(\text{Vect}(S^1)\big)^*$, under which the metric coefficients $V_1$ and $V_2$ would transform as coadjoint vectors [124].

### 4.3 Back to the Poincaré subgroup

Now that we have explored the whole algebraic and group structure of the extended Poincaré transformations, we are ready to come back to the original set of symmetry transformations, consisting in the Möbius transformations given by the vanishing Schwarzian time reparametrizations and the field translations with vanishing third derivatives. We would like to make explicit how they combine together to form the Poincaré group ISO(2, 1).

Starting with the proper time reparametrizations $T_f$ for $f$ a Möbius transformation of the proper time, i.e.

$$f_M(\tau) = \frac{a\tau + b}{c\tau + d}, \qquad \text{with} \qquad M = \begin{pmatrix} a & b \\ c & d \end{pmatrix}, \quad \det M = ad - bc = 1, \qquad (4.18)$$

the composition law for reparametrizations along the circle simply translates in the multiplication of 2×2 matrices as

$$T_{f_{M_1}} \circ T_{f_{M_2}} = T_{f_{M_1} \circ f_{M_2}} = T_{f_{M_1 M_2}}, \qquad \forall M_1, M_2 \in \mathrm{SL}(2, \mathbb{R}). \qquad (4.19)$$

Similarly, the action by conjugation of proper time reparametrizations on field translations gets encoded in matrix terms. For this, we represent the adjoint action of $D_f$ on $T_g$ as the action by conjugation of $\mathrm{SL}(2, \mathbb{R})$ matrices on real symmetric matrices, i.e.

$$D_{f^{-1}} \circ T_g \circ D_f = T_{(g \circ f)/\dot{f}} \qquad \Rightarrow \qquad D_{f_{M^{-1}}} \circ T_{g_Z} \circ D_{f_M} = T_{g_{M^t Z M}}, \qquad (4.20)$$

where the translation parameters have been repackaged into a 2×2 symmetric matrix

$$g_Z(\tau) = \alpha + \beta\tau + \gamma\tau^2, \qquad Z = \begin{pmatrix} \gamma & \frac{\beta}{2} \\ \frac{\beta}{2} & \alpha \end{pmatrix}, \qquad Z = Z^t. \qquad (4.21)$$

Then, we consider the subset of the group $\mathrm{BMS}_3 = \mathrm{Vect}(S^1) \rtimes \mathrm{Diff}(S^1)$ consisting in the pairs $(Z, M) := (T_{g_Z}, D_{f_M})$, which stands standing for the composed transformations $T_{g_Z} \circ D_{f_M}$. These form a Lie group with the group multiplication law[6]

$$(Z_1, M_1).(Z_2, M_2) = \left( Z_1 + (M_1^{-1})^t Z_2 M_1^{-1}, M_1 M_2 \right), \qquad (4.22)$$

which we recognize as expected as the Poincaré group $\mathrm{ISO}(2, 1) = \mathbb{R}^3 \rtimes \mathrm{SL}(2, \mathbb{R})$.

This concludes the proof that the symmetry transformations we have exhibited do form the ISO(2, 1) Poincaré group, which is best understood as a subgroup of the much larger group $\mathrm{BMS}_3 = \mathrm{Vect}(S^1) \rtimes \mathrm{Diff}(S^1)$. BMS transformations, or extended Poincaré transformations, are not strictly speaking symmetries since they induce variations of the action with non-vanishing bulk terms, but we will come back to the study of this point in future work.

## 5 Black hole evolution as trajectories on AdS$_2$

In this section we explore the geometrical consequences of the action of $\mathrm{SL}(2, \mathbb{R})$ on the black hole phase space. This is especially relevant since the Hamiltonian constraint $H_\mathrm{p}$ generating the evolution in time belongs to the $\mathfrak{sl}(2, \mathbb{R})$ Lie algebra formed by the CVH observables, and thus exponentiates to $\mathrm{SL}(2, \mathbb{R})$ transformations. It is therefore natural to look at the black hole

---

[6]If we were to choose the other ordering, namely $(Z, M) := D_{f_M} \circ T_{g_Z}$, the group operation would read

$$(Z_1, M_1).(Z_2, M_2) = (M_2^t Z_1 M_2 + Z_2, M_1 M_2),$$

which is the other standard version of the Poincaré group multiplication law.

evolution as trajectories for the three observables $C$, $V_2$ and $H = H_{\mathrm{p}} + L_0^2$, generating the $\mathfrak{sl}(2,\mathbb{R})$ Lie algebra. Since these observables are constrained by the Casimir relation $C^2 + HV_2 = B^2$, with $B$ a constant of the motion, this means looking at the black hole trajectory in terms of an SL$(2,\mathbb{R})$ flow on the AdS$_2$ hyperboloid.

This is similar to what has been worked out for FLRW cosmology in [37–39], where the evolution can be mapped to geodesics on the AdS$_2$ geometry. The main difference here is the shift between the Hamiltonian constraint $H_{\mathrm{p}}$ and the $\mathfrak{sl}(2,\mathbb{R})$ generator $H = H_{\mathrm{p}} + L_0^2$. This apparently small point has far-reaching consequences: it translates into a non-vanishing acceleration along the evolution curve, in such a way that the black hole trajectories are not AdS$_2$ geodesics as for FLRW cosmological trajectories, but are instead identified as *horocycles* on the hyperboloid. Horocycles are curves whose normal geodesics converge all asymptotically in the same direction.

We note that this geometrical description of the black hole phase space relies essentially on the $\mathfrak{sl}(2,\mathbb{R})$ part and variables of the Poincaré algebra which we have unraveled.

## 5.1 Algebraic structure of the AdS$_2$ geometry

### 5.1.1 The $\mathfrak{sl}(2,\mathbb{R})$ parametrization

The AdS$_2$ manifold is a 1-sheet hyperboloid, which can be defined through its embedding in 3-dimensional Minkowski space by the quadratic equation

$$-X_0^2 + X_1^2 + X_2^2 = \ell^2,\tag{5.1}$$

where $\ell$ is the AdS curvature radius. A useful parametrization is given by the light-cone variables[7]

$$X_0 = -\ell\,\frac{u_+ - u_-}{1 + u_+ u_-}\,,\qquad X_1 = \ell\,\frac{u_+ + u_-}{1 + u_+ u_-}\,,\qquad X_2 = \ell\,\frac{u_+ u_- - 1}{1 + u_+ u_-}\,.\tag{5.2}$$

Starting with the flat 3-dimensional line element, $\mathrm{d}s^2 = -\mathrm{d}X_0^2 + \mathrm{d}X_1^2 + \mathrm{d}X_2^2$, the induced metric on the AdS$_2$ hyperboloid is

$$\mathrm{d}s^2 = 4\ell^2\,\frac{\mathrm{d}u_+\,\mathrm{d}u_-}{(1 + u_+ u_-)^2}\,.\tag{5.3}$$

Our goal is to explain how the Kantowski–Sachs phase space can be described in terms of the AdS$_2$ geometry. In fact, we distinguish the two sectors of the phase space: on the one hand the $\mathfrak{sl}(2,\mathbb{R})$ Lie algebra formed by the observables $C$, $H$ and $V_2$, which encodes the dynamics of the configuration variable $V_2$ and its momentum $P_2$, and on the other hand the sector generate by $V_1$ and $P_1$. The interesting fact is that the latter sector only affects the dynamics of the first sector through the constant of the motion $B = P_1 V_1$, which gives the value of the quadratic Casimir operator of the $\mathfrak{sl}(2,\mathbb{R})$ algebra (2.23). This condition allows to write the $\mathfrak{sl}(2,\mathbb{R})$ generators in terms of the canonical pair $(V_2, P_2)$ plus the value of $B$, i.e.

$$j_z = \frac{1}{2\sqrt{2}}\left(2V_2 + 2P_2 B + V_2 P_2^2\right),\qquad k_x = \frac{1}{2\sqrt{2}}\left(2V_2 - 2P_2 B - V_2 P_2^2\right),\qquad k_y = -V_2 P_2 - B\,.\tag{5.4}$$

This parametrization automatically satisfies the quadratic Casimir condition $-j_z^2 + k_x^2 + k_y^2 = B^2$. Given the Lie algebra with the flat metric

$$\mathrm{d}s^2 = -\mathrm{d}j_z^2 + \mathrm{d}k_x^2 + \mathrm{d}k_y^2\,,\tag{5.5}$$

---

[7]This parametrization corresponds to the stereographic projection to the $X_2 = 0$ plane through the base point $\mathbf{p} = (0,0,1)$.

this Casimir equation gives back the AdS$_2$ hyperboloid. The change of coordinates to light-cone variables is then explicitly given by

$$u_- = \frac{\sqrt{2}V_2}{2B + V_2 P_2}, \qquad u_+ = -\frac{P_2}{\sqrt{2}}. \tag{5.6}$$

Finally, this allows to express the line element in terms of the canonical pair $(V_2, P_2)$ as

$$ds^2 = -2B\,dV_2\,dP_2 + V_2^2\,dP_2^2. \tag{5.7}$$

Let us now introduce the last ingredients we need in order to describe the trajectories, namely the group action on $\mathfrak{sl}(2, \mathbb{R})$.

### 5.1.2 Isometries of AdS$_2$ and the SL(2, $\mathbb{R}$) action

The closed $\mathfrak{sl}(2, \mathbb{R})$ algebra exponentiates to Killing flows on AdS$_2$ by means of SL(2, $\mathbb{R}$) group elements. To represent this in an explicit way, we use the 2-dimensional matrix representation of the $\mathfrak{sl}(2, \mathbb{R})$ Lie algebra and map the phase space to real matrices with vanishing trace of the form

$$M = \begin{pmatrix} k_y & k_x - j_z \\ k_x + j_z & -k_y \end{pmatrix}. \tag{5.8}$$

The Poisson structure on Minkowski space spanned by the Lie algebra vectors $\mathcal{J} = \{j_z, k_x, k_y\}$ in (2.27) can be written in a more compact manner as the matrix product

$$\{\eta \cdot \mathcal{J}, M\} = i(\eta \cdot \lambda)M - iM(\eta \cdot \lambda), \tag{5.9}$$

where the Minkowski scalar product is $\eta \cdot \mathcal{J} = -\eta_z j_z + \eta_x k_x + \eta_y k_y$, and the $\mathfrak{sl}(2, \mathbb{R})$ generators are represented by the matrices

$$\lambda_z = \frac{i}{2}\begin{pmatrix} 0 & 1 \\ -1 & 0 \end{pmatrix}, \qquad \lambda_x = \frac{i}{2}\begin{pmatrix} 0 & 1 \\ 1 & 0 \end{pmatrix}, \qquad \lambda_y = \frac{i}{2}\begin{pmatrix} 1 & 0 \\ 0 & -1 \end{pmatrix}. \tag{5.10}$$

This exponentiates to the adjoint representation of the SL(2, $\mathbb{R}$) Lie group on its algebra expressed as the matrix product

$$e^{\{\eta \cdot \mathcal{J}, \bullet\}}M = GMG^{-1}, \qquad \text{with} \qquad G = e^{i\eta \cdot \lambda} \in \text{SL}(2, \mathbb{R}). \tag{5.11}$$

The quadratic Casimir condition $\mathcal{C}_{\mathfrak{sl}(2,\mathbb{R})} = j_z^2 - k_x^2 - k_y^2 = -B^2$ simply becomes $\det(M) = -B^2$, which is manifestly invariant under the group action above. As a consequence, SL(2, $\mathbb{R}$) is indeed the group of isometries on AdS$_2$. These isometries take an elegant expression in terms of the null parametrization $u_\pm$. Indeed, the action of a general group element $G \in \text{SL}(2, \mathbb{R})$ on AdS$_2$ becomes a Möbius transformation on the null coordinates, i.e.

$$G = \begin{pmatrix} a & b \\ c & d \end{pmatrix} \in \text{SL}(2, \mathbb{R}), \qquad u_+ \mapsto G \triangleright u_+ = \frac{a\,u_+ + b}{c\,u_+ + d}, \qquad u_- \mapsto (G^{-1})^t \triangleright u_- = \frac{d\,u_- - c}{a - b\,u_-}. \tag{5.12}$$

## 5.2 Physical trajectories

We now look at the flow along the physical trajectories generated by the Hamiltonian constraint. As already pointed out in the section about the classical evolution, the $\mathfrak{sl}(2, \mathbb{R})$ generator $H$ differs from the Hamiltonian density $H_p$ by a constant shift $L_0^2$. This means that we can

recover the time evolution by the group flow, $e^{\{-\tau H_{\mathrm{P}},\bullet\}} = e^{\{-\tau H,\bullet\}}$, while we must nevertheless be careful to select the trajectories with a non-vanishing value $H = L_0^2 \neq 0$.

We easily integrate the SL$(2,\mathbb{R})$ flow generated by $H$ to find

$$M(\tau) = G_\tau M^{(0)} G_\tau^{-1}, \qquad \text{with} \qquad G_\tau = e^{i\frac{\tau}{\sqrt{2}}(\lambda_z - \lambda_x)} = \begin{pmatrix} 1 & 0 \\ \frac{\tau}{\sqrt{2}} & 1 \end{pmatrix}, \qquad (5.13)$$

which gives the evolution of the $\mathfrak{sl}(2,\mathbb{R})$ generators with respect to the proper time $\tau$ in the form

$$j_z(\tau) = j_z^{(0)} + \frac{\tau}{\sqrt{2}} k_y^{(0)} - \frac{\tau^2}{4}\big(k_x^{(0)} - j_z^{(0)}\big), \qquad (5.14a)$$

$$k_x(\tau) = k_x^{(0)} + \frac{\tau}{\sqrt{2}} k_y^{(0)} - \frac{\tau^2}{4}\big(k_x^{(0)} - j_z^{(0)}\big), \qquad (5.14b)$$

$$k_y(\tau) = k_y^{(0)} - \frac{\tau}{\sqrt{2}}\big(k_x^{(0)} - j_z^{(0)}\big). \qquad (5.14c)$$

Taking into account the on-shell condition $H = (k_x - j_z)/\sqrt{2} \approx L_0^2$ and enforcing the Casimir condition uniquely specifies the initial condition $M^{(0)}$ (up to a constant time shift) in the form

$$M^{(0)} = \begin{pmatrix} B & \sqrt{2}L_0^2 \\ 0 & -B \end{pmatrix}. \qquad (5.15)$$

Then, translating the evolution for the $\mathfrak{sl}(2,\mathbb{R})$ generators into the original canonical pairs $(V_i, P_i)$ as given by the parametrization (5.4), we can check that this is in perfect agreement with the classical trajectories (2.15) found in the previous sections. In figure 2, we draw the physical trajectories on the AdS$_2$ hyperboloid for a fixed value of the constant of the motion $B = 1$.

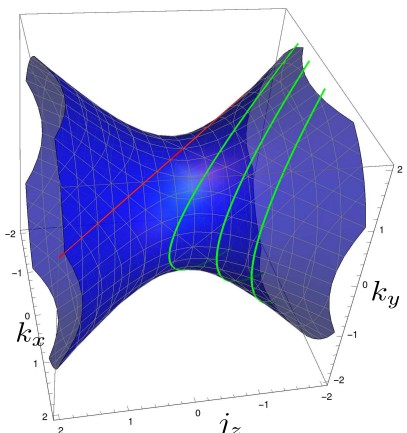

Figure 2: Plot of the physical trajectories corresponding to the flow generated by $H$ on a fixed hyperboloid, for $B = 1$. For non-vanishing $L_0$, these are horocycles on the AdS hyperboloid. The green curves correspond to $L_0 = 0.5, 1, 1.5$. Only the case $L_0 = 0$ in red gives a (null) geodesic, similarly to the trajectories of flat FLRW cosmology.

An important feature of the classical trajectories is that, unlike for FLRW cosmology, they are not AdS$_2$ geodesics but AdS$_2$ *horocycles*. In the $u_\pm$ parametrization, the Hamiltonian flow (5.14) is represented by a hyperbolic curve

$$u_+\left(u_- - \frac{\sqrt{2}B}{L_0^2}\right) = -1, \qquad u_+ = \frac{\sqrt{2}}{\tau}, \qquad u_- = \frac{2B - L_0^2\tau}{\sqrt{2}L_0^2}. \qquad (5.16)$$

One can compute the acceleration along these curves and realize that it is non-vanishing and actually orthogonal to the velocity: these are not geodesics, but the orthogonal notion in some sense. Note that the way in which $L_0$ appears can be traced back to the contribution of the 3d Ricci scalar to the Hamiltonian.

Now, we would finally like to show that all the geodesics perpendicular to one of these curves converge to the same point $(u_+, u_-) = (0, \infty)$. For this, we first notice that the vector parallel to the trajectories has a positive norm $ds/d\tau = L_0^2$, so that all the perpendicular geodesics are tachyonic. Moreover, we have that the geodesic that intersects the trajectory at the point $(u_+, u_-) = \left( \frac{\sqrt{2}L_0^2}{B}, \frac{B}{\sqrt{2}L_0^2} \right)$, corresponding to $\tau = B/L_0^2$, is generated as a group flow by $\exp(i\eta/B\lambda_y)$ acting on the intersection point[8], with $\eta$ the proper time along the geodesic. Using the property that isometries map geodesics to geodesics, and the fact that the physical solutions are generated as a group flow, we obtain the class of geodesics perpendicular to the classical trajectories as

$$
\begin{aligned}
(u_+, u_-) &= e^{i\frac{\tau - B/L_0^2}{\sqrt{2}}(\lambda_z - \lambda_x)} e^{i\eta/B\lambda_y} \triangleright \left( \frac{\sqrt{2}L_0^2}{B}, \frac{B}{\sqrt{2}L_0^2} \right) \\
&= \left( \frac{\sqrt{2}L_0^2}{-B + Be^{\eta/B} + L_0^2\tau}, \frac{B + Be^{\eta/B} - L_0^2\tau}{\sqrt{2}L_0^2} \right).
\end{aligned}
\tag{5.17}
$$

We see that these geodesics indeed all converge to $(u_+, u_-) = (0, \infty)$ when $\eta \to \infty$. This point corresponds to $(j_z, k_x, k_y) \to (\infty, \infty, 0)$. This result is illustrated below on figure 3.

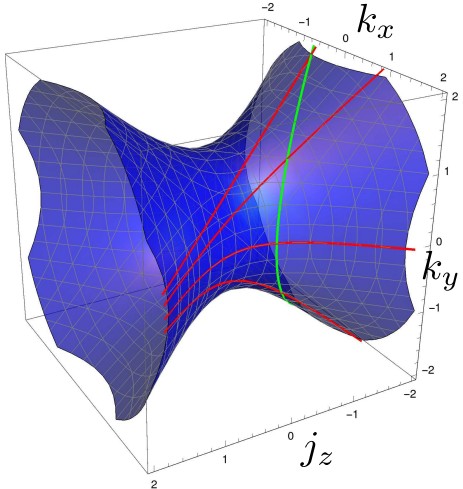

Figure 3: Plot of the perpendicular geodesics (in red) to the trajectory corresponding to $L_0 = 1$ (in green) for a Casimir value of $B = 1$. We see that the geodesics converge as $(j_z, k_x, k_y) \to (\infty, \infty, 0)$.

---

[8]This is a consequence of the fact that the group elements generate geodesics on $\text{AdS}_2$ when the corresponding generator vanishes on the flow lines. In this particular case we have that the group element $\exp(i\eta/B\lambda_y)$ generates a tachyonic geodesic on the plane $k_y = 0$, given by

$$
(u_+, u_-) = \left( \frac{\sqrt{2}L_0^2}{B} e^{-\eta/B}, \frac{B}{\sqrt{2}L_0^2} e^{\eta/B} \right).
$$

This geodesic is perpendicular to the trajectory and intersects it at $(u_+, u_-) = \left( \frac{\sqrt{2}L_0^2}{B}, \frac{B}{\sqrt{2}L_0^2} \right)$.

# 6 Symmetry-preserving regularization: solving the singularity

In this section we show that it is possible to define a "polymer" regularization for the black hole evolution, inspired by the techniques of loop quantum cosmology (LQC), while preserving the classical Poincaré symmetry which we have identified. This regularization resolves the black hole singularity and replaces it by a black-to-white hole transition. At the technical level, this is simply realized by a canonical transformation on the Kantowski–Sachs phase space, similarly to what was achieved for the polymer regularization of the big bang singularity of FLRW cosmology in [38].

The main ingredient of the polymer regularization scheme, as used in effective LQC [68], is to replace a suitable choice of phase space coordinates, say $q$ by their "polymerized" expression[9] $\sin(\lambda q)/\lambda$, where $\lambda$ is a UV regulator related to the minimal area gap of full loop quantum gravity. These heuristic corrections, which one can put by hand in order to study the effective classical dynamics, but would nevertheless like to *derive* from the full theory, are supposed to capture features of the quantization scheme used in loop quantum gravity. There, one is working with Yang–Mills like connection variables, which in the quantum theory are only represented as exponentiated holonomy operators. For this reason, the polymerisation is also known as the inclusion of holonomy corrections.

The form of this polymerization scheme (i.e. which variables to polymerize and which regulator to choose) has been heavily debated in the context of the Kantowski–Sachs black hole interior spacetime [72–94]. It is therefore natural to use the notion of symmetry to study this problem, and we have now a powerful criterion at our disposal. To start with, consider for example the polymerization of the momenta as

$$P_i \to \frac{\sin(\lambda_i P_i)}{\lambda_i} \qquad \forall\, i \in \{1,2\}, \qquad (6.1)$$

where the $\lambda_i$'s are effective parameters encoding the quantum gravity corrections, and in terms of which the effective Hamiltonian becomes

$$H_{\mathrm{p}}^{(\lambda)} = -L_0^2 - V_1 \frac{\sin(\lambda_1 P_1)}{\lambda_1} \frac{\sin(\lambda_2 P_2)}{\lambda_2} - V_2 \frac{\sin^2(\lambda_2 P_2)}{2\lambda_2^2}\,. \qquad (6.2)$$

Here the $\lambda_i$'s are taken to be constants in the phase space, which therefore represents the so-called $\mu_0$ scheme[10]. In the low curvature regime $\lambda_i P_i \ll 1$, the classical evolution is restored. However, defining the regularized shifted Hamiltonian density as $H^{(\lambda)} = H_{\mathrm{p}}^{(\lambda)} + L_0^2$, one can easily see that regularization breaks the Poincaré symmetry unraveled in the previous sections. While it is of course plausible that the quantization of a theory breaks a classical symmetry, with possible deep physical reasons for this, and consequences, here we take the viewpoint that there should actually exist a polymer regularization which preserves the $\mathfrak{iso}(2,1)$ Poincaré algebra and the symmetries it generates.

We recall in appendix B the CVH structure within the full algebra of constraints of LQG (which however does not have an interpretation in terms of symmetries of the action so far), and the link between the $(V_i, P_i)$ black hole phase space used here and the $(b, c, p_b, p_c)$ phase space variables traditionally used in LQC.

---

[9]One can more generally consider higher Fourier modes and arbitrary compactifying functions, as in [125].

[10]It would be possible, and interesting, to consider $\bar{\mu}$ schemes where the $\lambda_i$'s are functions of the phase space variables. In this case, contrary to the $\mu_0$ scheme, it is not guaranteed that it is always possible to realize the polymerization as a canonical transformation. Investigating this in details would allow to discriminate between the various polymerization (or $\mu$) schemes which have been proposed in the literature.

## 6.1 Polymerization as a canonical transformation

As pointed out in the context of FLRW LQC in [38], a systematic way to ensure that the Poisson bracket relations are preserved is to perform a canonical transformation on the phase space. It is possible to use such a redefinition of the phase space variables to produce a polymerized version of the Hamiltonian constraint and of the Poincaré generators. A general symplecto-morphism changing the canonical momenta is by definition

$$(V_i, P_i) \rightarrow (v_i, p_i) \qquad \text{with} \qquad \{v_i, p_j\} = \delta_{ij}, \tag{6.3}$$

with defining functions $F_i$ such that

$$P_i = F_i(p_i) \qquad V_i = \left(\frac{\mathrm{d}F_i}{\mathrm{d}p_i}\right)^{-1} v_i + k, \qquad k = \text{constant}. \tag{6.4}$$

The lowercase $(v_i, p_i)$ represent the new *polymer* variables obtained by transformation of the original phase space canonical pairs $(V_i, P_i)$.

Let us now find a canonical transformation leading to the Hamiltonian (6.2). For this, we start with the sector $(v_2, p_2)$. The effective Hamiltonian (6.2), now written in terms of lowercase variables, gives the following equations of motion:

$$\dot{v}_2 = \{v_2, H_{\mathrm{p}}\} = -\cos(\lambda_2 p_2)\left(v_1 \frac{\sin(\lambda_1 p_1)}{\lambda_1} + v_2 \frac{\sin(\lambda_2 p_2)}{\lambda_2}\right), \tag{6.5a}$$

$$\dot{p}_2 = \{p_2, H_{\mathrm{p}}\} = \frac{\sin^2(\lambda_2 p_2)}{2\lambda_2^2}. \tag{6.5b}$$

Let us solve the evolution by deparametrizing it. We recall that under the hypothesis that $H_{\mathrm{p}} = 0$ this procedure does not depend on the lapse. Using this constraint, the evolution equation for $v_2$ with respect to $p_2$ reads

$$\frac{\mathrm{d}v_2}{\mathrm{d}p_2} = -\lambda_2 \cot(\lambda_2 p_2)\left(v_2 - \frac{2L_0^2\lambda_2^2}{\sin^2(\lambda_2 p_2)}\right). \tag{6.6}$$

This differential equation then integrates to

$$v_2(p_2) = -\frac{2\lambda_2}{\sin(\lambda_2 p_2)}\left(B + \frac{L_0^2\lambda_2}{\sin(\lambda_2 p_2)}\right). \tag{6.7}$$

This can now be compared with the classical deparametrized evolution (2.15), which is

$$V_2(P_2) = -\frac{2}{P_2}\left(B + \frac{L_0^2}{P_2}\right), \tag{6.8}$$

to find that the mapping (6.3) between the classical variables and the polymerized ones must satisfy

$$\frac{2}{F_2}\left(B + \frac{L_0^2}{F_2}\right) = \frac{2\lambda_2}{\sin(\lambda_2 p_2)}\left(B + \frac{L_0^2\lambda_2}{\sin(\lambda_2 p_2)}\right)\left(\frac{\mathrm{d}F_2}{\mathrm{d}p_2}\right)^{-1} - k. \tag{6.9}$$

Requiring the canonical transformation $F$ to be independent from the initial condition $B$, we need to impose

$$\frac{2}{F_2} = \frac{2\lambda_2}{\sin(\lambda_2 p_2)}\left(\frac{\mathrm{d}F_2}{\mathrm{d}p_2}\right)^{-1} \qquad \Rightarrow \qquad F_2(p_2) = \frac{2}{\lambda_2}\tan\left(\frac{\lambda_2 p_2}{2}\right), \tag{6.10}$$

and fix the value of the constant to $k = \lambda_2^2 L_0^2/2$. At the end of the day, this gives the following canonical transformation:

$$P_2 = \frac{2}{\lambda_2} \tan\left(\frac{\lambda_2 p_2}{2}\right), \qquad V_2 = v_2 \cos^2\left(\frac{\lambda_2 p_2}{2}\right) + \frac{\lambda_2^2 L_0^2}{2}. \qquad (6.11)$$

Following the same procedure for the sector $(v_1, p_1)$, we find

$$P_1 = \frac{2}{\lambda_1} \tan\left(\frac{\lambda_1 p_1}{2}\right), \qquad V_1 = v_1 \cos^2\left(\frac{\lambda_1 p_1}{2}\right). \qquad (6.12)$$

The change of variables defined by (6.11) and (6.12) preserves the Poisson brackets by construction, and therefore the phase space Poincaré symmetry. We can thus obtain the *polymerized* Hamiltonian (up to a redefinition of the lapse) as well as regularized versions of all the other observables $A, C, D$ such that the $\mathfrak{iso}(2,1)$ algebra is preserved. In particular, the regularized effective Hamiltonian reads[11]

$$H^{(\lambda)} = -\cos^{-2}\left(\frac{\lambda_2 p_2}{2}\right)\left(v_1 \frac{\sin(\lambda_2 p_2)}{\lambda_2}\frac{\sin(\lambda_1 p_1)}{\lambda_1} + v_2 \frac{\sin^2(\lambda_2 p_2)}{2\lambda_2^2}\right). \qquad (6.13)$$

Having shown that this regularization preserves the Poincaré symmetry, we now show that it also solves the black hole singularity and replaces it by a black-to-white hole transition.

## 6.2 Evolution of the polymerized model

The key step leading to the singularity resolution in the effective model is to replace the variables in the metric coefficients by the polymerized one. Moreover, in order to be in line with the existing literature on effective polymer Hamiltonian dynamics in cosmology, we will reabsorb the lapse in a redefinition of the proper time in the metric. The effective line element which we consider is therefore

$$ds^2 = -\frac{v_1 L_0^2}{2v_2}d\tau'^2 + \frac{8v_2}{v_1}dx^2 + v_1 d\Omega^2, \qquad d\tau = \cos^2\left(\frac{\lambda_2 p_2}{2}\right)d\tau'. \qquad (6.14)$$

We can then invert the canonical transformation and find the evolution of the new variables. In order to see if the singularity is resolved, we need to find out if $v_1$ is vanishing at some point. We have that

$$v_1 = V_1\left(1 + \frac{\lambda_1^2 P_1^2}{4}\right) = V_1 + \frac{\lambda_1^2 B^2}{4V_1}. \qquad (6.15)$$

It is evident from this expression that, although $V_1$ follows the classical trajectory towards the singularity at $V_1 = 0$, the modified variable $v_1$ never vanishes and reaches a minimum at

$$v_1^{(\mathrm{min})} = \lambda_1 B. \qquad (6.16)$$

---

[11]The other regularized Poincaré observables are

$$A = \frac{2}{\lambda_2^2}v_1 \tan^2\left(\frac{\lambda_2 p_2}{2}\right)\cos^2\left(\frac{\lambda_1 p_1}{2}\right),$$

$$C = \frac{\sin(\lambda_1 p_1)}{\lambda_1}v_1 + \frac{\sin(\lambda_2 p_2)}{\lambda_2}v_2 + \lambda_2 \tan\left(\frac{\lambda_2 p_2}{2}\right),$$

$$D = \frac{2}{\lambda_2}v_1 \tan\left(\frac{\lambda_2 p_2}{2}\right)\cos^2\left(\frac{\lambda_1 p_1}{2}\right).$$

The explicit time evolution for the two configuration variables is given by

$$v_1 = \frac{A}{2}\tau^2 + \frac{\lambda_1^2 B^2}{2A\tau^2}, \tag{6.17a}$$

$$v_2 = B\left(\tau + \tau^{-1}\lambda_2^2\right) - \frac{L_0^2}{2}\left(\tau + \tau^{-1}\lambda_2^2\right)^2, \tag{6.17b}$$

which can be compared with the classical evolution (2.15) recovered in the limit $\lambda_i \to 0$. Note that these effective evolution equations are much simpler than the evolution equations obtained in the usual polymerization schemes [85, 87–90, 126]. This effective bouncing evolution is represented on figure 4 below.

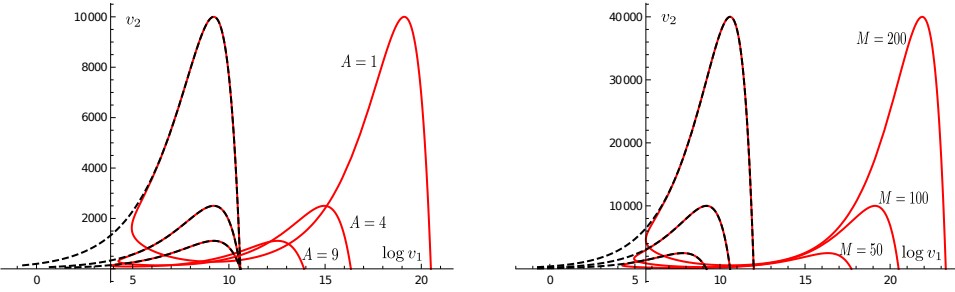

Figure 4: Plot of the *effective* trajectories in configuration space $(\log v_1, v_2)$ in solid red, compared to the respective classical ones in dashed. On the left, the value of the BH mass (i.e. $B\sqrt{A}$) is fixed to $M = 100$ and $A = 1, 4, 9$. On the right, the mass is varied as $M = 50$, 100, 200, and $A = 1$. We take both $\lambda_1 = \lambda_2 = 1$, as well as $L_0 = 1$.

Note that the *on-shell* value of the line element (6.14) is not a solution of the Einstein equations because of the (effective) quantum effects. On the other hand, we expect these effect to be negligible in the low curvature regime, which here corresponds to the region near the black hole horizon. This is when $\tau \to 2B$, where we have

$$v_1 \sim \frac{A}{2}\tau^2, \qquad v_2 \sim B\tau - \frac{L_0^2}{2}\tau^2. \tag{6.18}$$

Repeating the change of variables (2.16) of the previous section, we find a Schwarzschild metric with $M_{\text{BH}} = B\sqrt{A}/\sqrt{2}$. This shows that near the black hole horizon there are no corrections at leading order. The quantum effects becomes important as we approach the would-be singularity, which is at $\tau \to 0$ and gives

$$v_1 \sim \frac{B^2\lambda_1^2}{2A\tau^2} + \mathcal{O}(t^2), \qquad v_2 \sim -\lambda_2^2\left(L_0^2 + \frac{L_0^2\lambda_2^2}{2\tau^2} - \frac{B}{\tau}\right) + \mathcal{O}(t). \tag{6.19}$$

If we now set

$$\tau = \frac{B\lambda_1}{\sqrt{2A}}\frac{1}{t}, \qquad x = \frac{B\lambda_1}{2L_0\sqrt{2A}\lambda_2^2}r, \tag{6.20}$$

the metric (6.14) becomes

$$\mathrm{d}s^2 = -f(t)^{-1}\mathrm{d}t^2 + f(t)\mathrm{d}r^2 + t^2\mathrm{d}\Omega^2, \tag{6.21}$$

with

$$f(t) = \left(\frac{2M_{\text{WH}}}{t} - 1\right), \qquad M_{\text{WH}} = \frac{B^2\lambda_1}{\lambda_2^2 L_0^2\sqrt{2A}}. \tag{6.22}$$

Just like in most of the effective studies of the LQC black hole interior, we find that the singularity is replaced by a black-to-white hole transition. The physical radius represented by $\sqrt{v_1}$ is never vanishing and reaches a minimal value (6.16), where we have a minimal sphere which is a *transition surface* between the trapped (BH) and anti-trapped (WH) space-like regions. This can be checked by computing the expansion of the future pointing null normal to the 2-sphere at constant time, which changes sign while passing through the transition surface. We refer the reader to [88] for details about the causal structure of this kind of effective spacetimes.

Let us conclude with a comment on the role of the fiducial length $L_0$. As pointed out in the first section, under a rescaling $L_0 \to \alpha L_0$ the momenta scale linearly with $\alpha$, which implies that $B$ scales linearly while $A$ scales quadratically. In order to have a well-defined regularization (6.1), we also need to rescale the UV regulator $\lambda_i \to \alpha^{-1}\lambda_i$ so that the product $P_i\lambda_i$ is invariant. Altogether, we find that the expression for the white hole mass is therefore unaffected by a scaling of the cell.

# 7 Discussion and perspectives

In this article we have revealed surprising features inside a simple yet physically relevant system in general relativity, namely the black hole interior spacetime described by a Kantowski–Sachs family of metrics. Because of the homogeneity, the Einstein–Hilbert action for these metrics (2.2) reduces to a mechanical model whose configuration space consists of two degrees of freedom. Its phase space dynamics is well-known and can be explicitly solved. Yet, in the line of recent work on homogeneous FLRW cosmology [37–40], we have unraveled an $\mathfrak{iso}(2,1)$ algebra of observables on this phase space. This Poincaré structure has the interesting property of controlling the dynamics and, perhaps more intriguingly, of lifting up to Lagrangian symmetries. These symmetries decompose into a conformal $\mathfrak{sl}(2,\mathbb{R})$ sector acting as Möbius transformations of the proper time, and a 3-dimensional Abelian part generating translations of one of the configuration variables.

In section 3, we have studied in details the Poincaré symmetries of the classical action. We stress out once again that these are not mere time reparametrizations left over from diffeomorphism invariance by the fixing to homogeneous metrics. These are *new symmetries* existing on top. We have computed their conserved Noether charges, and found that they are actually the initial conditions for the evolution of the phase space functions forming the above mentioned Poincaré algebra. The $\mathfrak{iso}(2,1)$ algebra of observables on the phase space is therefore a dynamical, time evolved version of the Noether charges. Importantly, we have computed the action of the symmetries of the Lagrangian on the physical trajectories of the system, and found that they act indeed as *physical symmetries* changing the mass of the black hole.

Perhaps even more surprisingly, we have revealed in section 4 that the newly discovered Poincaré symmetries actually descend from $\text{BMS}_3$ transformations. Although general BMS transformations are not symmetries of the action, this has revealed that the configuration space variables $V_1$ and $V_2$ have a natural interpretation as $\mathfrak{bms}_3$ Lie algebra elements. Although the role of BMS symmetries in the context of black hole spacetimes has been considered now for a long time, it is the first time, to our knowledge, that such hints at a BMS structure appears in mini-superspace descriptions of the black hole interior. This raises many interesting questions, such as that of the dimensionality: why is it $\text{BMS}_3$ which appears in this 4-dimensional context? Naturally, one could think that this is because of the symmetry reduction to a cosmological model, but the question remains open. Then, this also begs the question of whether boundaries should play a role in our construction. Indeed, boundaries (asymptotic or at finite distance) are the natural location where symmetries such as BMS live, and in the homogeneous black hole interior they could play a role which has been obscured by the symmetry reduction. This

is also supported by the intriguing role of the IR cutoff $L_0$ which we have witnessed at various stages of our study. It seems that the next step ahead is to unleash some freedom for this cutoff, to let the boundary evolve in time, and to discuss more precisely the variational principle and the role of the boundary terms. We have started to investigate these aspects.

Going back to more pragmatic grounds, we have used in section 5 the availability of an AdS$_2$ structure on phase space to reformulate the dynamics geometrically. This gives rise to an interesting picture where physical trajectories are horocycles on the hyperboloid.

We have proposed to use this new way of looking at the black hole interior phase space to study the quantum theory. Ideally, and as a natural next step, we are going to use Poincaré representation theory to bring the algebraic and geometrical structures on phase space into the quantum realm. This can be done because we have $\mathfrak{sl}(2,\mathbb{R})$ and Poincaré Casimir balance relations guiding the representation theory. This is reminiscent of how quantization of (boundary) symmetry algebras has been proposed in [25, 127].

In lack of an explicit quantization for the moment, we have studied the effective dynamics encoding heuristic quantum gravity corrections, in the spirit of effective loop quantum cosmology. Indeed, it is also very natural to use symmetry principles as a non-perturbative guide to reduce quantization ambiguities. Such quantization ambiguities have precisely been discussed recently in the study of the black hole interior in LQC. Here we have shown in section 6, following [103], that it is possible to view the polymerization as a canonical transformation on phase space. This has the advantage of intertwining, by construction, any structure on phase space such as the symmetry discovered here. We have exhibited such a canonical transformation to polymerized variables, and shown that in terms of these variables the evolution of the black hole interior is non-singular and results in a replacement of the singularity by a black-to-white hole transition. For future work it will be interesting to study more quantitatively this non-singular evolution, and also to perform the quantization of the model along the lines mentioned above.

Finally, let us point out that a clear understanding of the origin of the Poincaré symmetry, and its possible relationship with the spacetime BMS symmetries, could be achieved by studying the symmetries of the inhomogeneous black hole (interior and exterior) spacetime. It is possible to consider for example a spherically-symmetric ansatz for the metric, and thereby reduce the problem to a two-dimensional $(r, t)$-plane gravitational theory with an inhomogeneous radial direction. Upon imposition of homogeneity this reduces to the Kantowski–Sachs model studied here, and for which the symmetries have been identified. The question is then that of the origin and the realization of these symmetries in the inhomogeneous precursor model. We are currently investigating this problem.

## A   List of variables

In this appendix we gather some of the notations which are used throughout the paper to denote certain variables. In particular, the Poincaré generators have appeared in many different forms depending on the context. We give here these different forms as well as the equations defining them.

## B   Connection-triad variables

Early work on the effective dynamics of the Kantowski–Sachs black hole interior used a slightly different language based on the connection-triad variables of loop quantum gravity [69, 75]. In this appendix write down the relationship between these and our variables, and also recall

| phase space variables | coordinates $(V_1, V_2)$ momenta $(P_1, P_2)$ |
|---|---|
| phase space $\mathfrak{iso}(2,1)$ generators | $\mathfrak{so}(2,1)$ part $(V_2, C, H)$ translation part $(V_1, A, D)$ |
| algebraic $\mathfrak{iso}(2,1)$ generators defined as (2.27) | $\mathfrak{so}(2,1)$ part $(j_z, k_x, k_y)$ translation part $(\Pi_0, \Pi_x, \Pi_y)$ |
| Noether charge $\mathfrak{iso}(2,1)$ generators defined as (3.12) and (3.15) | $\mathfrak{so}(2,1)$ part $(Q_0, Q_-, Q_+)$ translation part $(P_0, P_-, P_+)$ |

the structure of the CVH algebra of full LQG.

The line element used in e.g. [75] is

$$
\mathrm{d}s^2 = -N^2 \mathrm{d}t^2 + \frac{p_b^2}{L_0^2 p_c} \mathrm{d}x^2 + p_c \mathrm{d}\Omega^2 \, .
\tag{B.1}
$$

This corresponds to the choice of a densitized triad

$$
E = E_i^a \tau^i \partial_a = p_c \sin\theta \, \tau_1 \partial_x + \frac{p_b}{L_0} \sin\theta \, \tau_2 \partial_\theta + \frac{p_b}{L_0} \tau_3 \partial_\phi \, .
\tag{B.2}
$$

In LQG this densitized triad is canonically conjugated to the SU(2) Ashtekar–Barbero connection, and the symplectic structure is

$$
\{A_a^i(x), E_j^b(y)\} = 8\pi\gamma \delta_a^b \delta_j^i \delta(x - y) \, .
\tag{B.3}
$$

In the case of the homogeneous spherically-symmetric BH interior the connection is

$$
A = A_a^i \tau_i \mathrm{d}x^a = \frac{c}{L_0} \tau_1 \mathrm{d}x + b\tau_2 \mathrm{d}\theta + b \sin\theta \, \tau_3 \mathrm{d}\phi + \cos\theta \, \tau_1 \mathrm{d}\phi \, ,
\tag{B.4}
$$

and the Poisson brackets reduce to

$$
\{c, p_c\} = 2\gamma \qquad\qquad \{b, p_b\} = \gamma \, .
\tag{B.5}
$$

These phase space variables are related to the ones used here by

$$
V_1 = p_c \, , \qquad V_2 = \frac{1}{8} \frac{p_b^2}{L_0^2} \, , \qquad P_1 = -\frac{1}{2\gamma} c \, , \qquad P_2 = -\frac{4L_0^2}{\gamma} \frac{b}{p_b} \, .
\tag{B.6}
$$

We now close this appendix by commenting on the CVH algebra structure of full general relativity in Ashtekar–Barbero canonical variables. This follows the appendix of [37] and corrects a minor point there. We then connect the results to the spherically-symmetric homogeneous spacetime considered here. The connection-triad variables are built from the spatial triad $e_a^i \mathrm{d}x^a$ and the extrinsic curvature $K_a^i \mathrm{d}x^a$, where $i, j, k$ are internal $\mathfrak{su}(2)$ indices, as

$$
E_i^a = \det(e_a^i) e_i^a \, , \qquad A_a^i = \Gamma_a^i[E] + \gamma K_a^i \, ,
\tag{B.7}
$$

with the torsion-free spin connection $\Gamma_a^i[E]$ is

$$
\Gamma_a^i = \frac{1}{2} \epsilon^{ijk} E_k^b \left( 2\partial_{[b} E_{a]}^j + E_j^c E_a^l \partial_b E_c^l \right) + \frac{1}{4} \frac{\epsilon^{ijk} E_k^b}{\det E} \left( 2E_a^j \partial_b \det E - E_b^j \partial_a \det E \right) .
\tag{B.8}
$$

The canonical pairs $(A^i_a, E^i_a)$ are subject to seven first class constraints given by Gauss, diffeomorphism and scalar constraints. Here we are interested in the scalar constraint

$$H[N] = \int_\Sigma d^3x\, \mathcal{H} = \frac{1}{16\pi} \int_\Sigma d^3x\, \frac{N}{\sqrt{q}} E^a_i E^b_j \left( \varepsilon^{ij}{}_k F^k_{ab} - 2(1+\gamma^2) K^i_{[a} K^j_{b]} \right). \qquad (B.9)$$

It is useful to split the constraint into the so-called Euclidean $H_{\mathrm{E}}$ and Lorentzian $H_{\mathrm{K}}$ parts given by

$$H_{\mathrm{E}}[N] = \int_\Sigma d^3x\, N \mathcal{H}_{\mathrm{E}} = \frac{1}{16\pi} \int_\Sigma d^3x\, N \frac{E^a_i E^b_j}{\sqrt{\det E}} \left( \varepsilon^{ij}{}_k F^k_{ab} \right), \qquad (B.10a)$$

$$H_{\mathrm{K}}[N] = \int_\Sigma d^3x\, N \mathcal{H}_{\mathrm{K}} = -\frac{1+\gamma^2}{8\pi} \int_\Sigma d^3x\, N \frac{E^a_i E^b_j}{\sqrt{\det E}} K^i_{[a} K^j_{b]}. \qquad (B.10b)$$

The generator of dilation in the phase space is here the trace of the extrinsic curvature, also known as the complexifier (hence its name $C$ in the main text) as it plays a primary role in defining the Wick transform between real and self-dual version of LQG. The last quantity to consider is the volume of the space-like hypersurface. These quantities are given by

$$C = \frac{1}{8\pi} \int_\Sigma d^3x\, E^a_i K^i_a, \qquad V = \int_\Sigma d^3x\, \sqrt{\det(E^a_i)}. \qquad (B.11)$$

A straightforward calculation shows that, together with the Lorentzian part for a constant lapse function, they form an $\mathfrak{sl}(2,\mathbb{R})$ CVH algebra

$$\{C, V\} = \frac{3}{2} V, \qquad \{V, H_{\mathrm{K}}\} = (1+\gamma^2) 8\pi C, \qquad \{C, H_{\mathrm{K}}\} = -\frac{3}{2} H_{\mathrm{K}}. \qquad (B.12)$$

On the other hand, if we also consider the Euclidean part of the Hamiltonian constraint the algebra fails to be closed and we find

$$\{V, H_{\mathrm{E}}\} = -8\pi\gamma^2 C, \qquad \{C, H_{\mathrm{E}}\} = \frac{1}{2} H_{\mathrm{E}} + 2 \frac{\gamma^2}{1+\gamma^2} H_{\mathrm{K}}, \qquad (B.13)$$

$$\{H_{\mathrm{E}}, H_{\mathrm{K}}\} = \frac{1+\gamma^2}{16\pi} \int_\Sigma d^3x \left[ \epsilon^{abc} F^i_{bc} K^i_a - 6\gamma^2 \det(K^i_a) \right].$$

Note the particular role played by the self-dual value $\gamma = \pm i$.

In flat FLRW cosmology, it turns out that the Euclidean and Lorentzian parts of the Hamiltonian constraint are actually proportional. This is however not the case for the Kantowski–Sachs geometry, where with the variables introduced above and for $N = 1$ we get

$$H_{\mathrm{E}} = \frac{2bc p_c + (b^2-1) p_b}{2\sqrt{p_c}}, \qquad H_{\mathrm{K}} = -(1+\gamma^{-2}) \frac{2bc p_c + b^2 p_b}{2\sqrt{p_c}}, \qquad (B.14)$$

while the complexifier and the volume are given by

$$C = \frac{2b p_b + c p_c}{2\gamma}, \qquad V = 4\pi p_b \sqrt{p_c}. \qquad (B.15)$$

The algebra is then obviously given again by (B.12) and (B.13), with the last bracket changed to

$$\{H_{\mathrm{E}}, H_{\mathrm{K}}\} = -\frac{1+\gamma^2}{2\gamma} c. \qquad (B.16)$$

The key idea which leads to the CVH algebra for the black hole interior is to change the lapse so as to recover the same property as in FLRW, namely a simple phase space independent relationship between $H_E$ and $H_K$. This choice corresponds to

$$N = \frac{2\sqrt{p_c}}{p_b} \quad \Rightarrow \quad H_K[N] = -(1+\gamma^2)(H_E[N]+1). \tag{B.17}$$

Using (B.6) shows that this lapse is indeed (up to a factor of $L_0$) the one used in (2.7). Re-absorbing the lapse and redefining the volume and the complexifier, we see that the modified CVH algebra gives indeed the $\mathfrak{sl}(2,\mathbb{R})$ sector of the $\mathfrak{iso}(2,1)$ structure presented in the main text, with

$$\frac{1}{1+\gamma^2}H_K[N] = -\frac{1}{2}(2P_1V_1 + P_2V_2)P_2 = H, \tag{B.18a}$$

$$V \rightarrow \frac{V}{16\pi N} = \frac{p_b^2}{8} \propto V_2, \tag{B.18b}$$

$$C \rightarrow C = \{V_2, H\} = -V_1P_1 - V_2P_2. \tag{B.18c}$$

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
