# Peer review of "Symmetries of the Black Hole Interior and Singularity Regularization"

_SciPost Physics, doi:SciPost Phys. 10, 022 (2021)_

## Round 1 · Referee Report · Anonymous · 2020-11-12

Strengths
1. The paper describes an essentially elementary, but important, physical problem from a fresh perspective that elegantly uses symmetries.
2. These symmetries turn out to be the 3D Poincaré group and may thus be related to BMS symmetry, suggesting connections with a very active current research trend.
3. The paper is written in a very clear and pedagogical manner.
4. Several natural follow-ups are listed, and seem both interesting and reasonably within reach.
Weaknesses
1. A strong technical limitation is imposed by the finite-dimensional, mini-superspace description of gravitational dynamics. This is part of the assumptions of the paper, so it is an unavoidable feature (and it is indeed presented as such by the authors), but it may limit the validity of the paper's conclusions. In particular, it is unclear whether Poincaré symmetry survives a looser phase space approach; addressing this question is mentioned by the authors as a follow-up, which seems quite reasonable.
Note that this drawback is the reason I'm judging the paper's significance as "good" instead of "high" below: a "high" would have required more generality, in my opinion.
2. The paper is somewhat lengthy, though this is understandable given the pedagogical effort invested by the authors.
Report
The paper addresses an important conceptual issue (namely the dynamics of black hole interiors) from a modern perspective inspired by asymptotic symmetries and near-AdS holography, and presents the problem in a very pedagogical way. Accordingly, it definitely deserves publication in SciPost Physics.
Some minor modifications are nevertheless required: see below.
Requested changes
1. The Hamiltonian is one of the generators of the non-Abelian sl(2,R) subalgebra of the Poincaré algebra. This should be contrasted with the usual space-time interpretation of the Poincaré algebra, where the Hamiltonian is one of the *Abelian* generators. This detail may have important implications for quantization and representation theory, so it should be stressed at some point. It is somewhat similar to the fact that the generator of time translations in Galilean CFTs is in fact a generator of rotations from the perspective of BMS symmetry (and vice-versa).
2. In section 4.2, the authors point out that the metric components $V_1$, $V_2$ transform under the adjoint representation of BMS$_3$. This is an important but surprising conclusion: in both 2D and 3D gravity, components of the metric rather transform under the *coadjoint* representation of the asymptotic symmetry group, which allows one to use the Lie-Poisson bracket in order to define symplectic structures on orbits. (In fact, this is even a general result on symmetries in symplectic geometry.) The *adjoint* space, by contrast, does not admit any standard Poisson structure. This should be mentioned, since it suggests that the BMS generalization of the setup studied by the authors does not admit a simple phase space interpretation, so that quantization may be problematic.
(A comment to the authors: it may be possible to solve this issue and turn the metric components into *coadjoint* vectors, at the expense of sacrificing BMS$_3$ symmetry in favour of its much less studied "dual group", namely Diff$\,S^1\ltimes($Vect$\,S^1)^*$. The latter also admits a Poincaré subalgebra, so the main conclusions would not change drastically, but it might lead to a better-behaved phase space interpretation. This dual group is mentioned in arXiv:1505.02031.)
We thank the referee for her/his constructive comments and suggestions for improvement. Here is a detailed reply to the questions/remarks raised by the referee.
- We have added a paragraph at the end of section 2 to clarify this point.
- We have added a paragraph at the end of section 4.2 to clarify this point. There is also a new paragraph in the introduction of section 4, mentioning that the BMS group studied in this work is different from the usual spacetime BMS group appearing in e.g. the study of asymptotic symmetries.
(in reply to Report 3 on 2020-12-16)
We thank the referee for her/his constructive comments and suggestions for improvement. Here is a detailed reply to the questions/remarks raised by the referee.

---

## Round 1 · Referee Report · Anonymous · 2020-12-15

Strengths
1. The quantization of the interior of black holes is an interesting topic that has recently received lots of attention from various communities. This paper sheds new light on this topic.
2. The results are thorough and extremely clearly presented.
Weaknesses
I am more skeptical of the claim that there should "definitely be a BMS group reformulation of the Kantowski-Sachs spacetime geometry". What is wrong with the following reasoning (which admittedly is not a perfect analogy, but seems to capture the essence well)? In any homogeneous model, the symmetry group includes translations. Since the BMS group also has translations as a subgroup, it seems that there is something "deep" here and I may reformulate my original set-up using the BMS group instead. However, we know that there is no BMS group in cosmology (for the case with positive cosmological constant [e.g., arXiv:1009.4730v1] and even in the case with a future null asymptote [arXiv:2009.01243]). It would be helpful if the authors substantiated their claim with additional arguments.
Report
This paper studies the interior of a Schwarzschild black hole from a symmetry perspective and unravels the existence of an iso(2,1) Poincaré algebra. The dynamics are carefully studied from the perspective of Mobius transformations of the proper time and translations. Finally, these classical results are used to reduce quantization ambiguities and study a polymer-type quantization of the interior showing a resolution of the singularity.
Requested changes
Suggestions
1. While the paper is very clearly written and the results are explained well, the paper would benefit from a diagram with the (many) variables used, their most important properties and the relation between them.
2. In the captions of the figures, the ordering of the different curves should be stated explicitly (e.g. in fig 1: which curve corresponds to which A/M).
3. In the introduction, already state what a horocycle is (as this is not a common curve).
4. In the introduction, it would be helpful for the reader to understand if the "symmetry" perspective advocated in this paper also removes quantization ambiguities in LQC and if so which ones.
5. After equation (2.15), it would be good to stress that despite having chosen to work with a more general line element in Eq. (2.2), these equations indicate that the solution space describes the line element in Eq. (2.1) and the symmetry properties are thus exactly those of the interior of a Schwarzschild black hole and not some more general metric.
6. In Eq. (3.10) please indicate which symmetry transformation generates which charge.
7. The spacetime signature in Sec. 2-4 is mostly plus, while in Sec. 5 it is changed to mostly minus. Please explain why, or change it such that it is consistent throughout.
8. In the caption of Fig. 3, it would be helpful if (u_+, u_-)=(0,\infty) would also be stated in terms of the values of k_x, k_y and j_z.
(9. Very minor: throughout the paper "consists in" --> "consists of".)
We thank the referee for her/his constructive comments and suggestions for improvement. Here is a detailed reply to the questions/remarks raised by the referee.
Let us start by answering the question raised by the referee in the section ’weaknesses’ of the report. Our remark that there should ‘definitely be a BMS group reformulation of the Kantowski-Sachs spacetime geometry’ is simply based on the results of section 4, which we believe are substantial evidence for the claim. In upcoming work we will indeed show that it is possible to reformulate the symmetry-reduced action in terms of a geometrical BMS group action. We indeed agree with the referee, and do not claim that in any homogeneous model, because of the presence of a translational symmetry, there should be a relationship with a BMS symmetry of some sort. This is indeed not true in FLRW cosmology for example, where the BMS structure which we have unraveled does not exist. The reason is simply that the translation part of the BMS group of section 4 is in fact unrelated to the spacetime translations of the Poincar\’e or ‘usual’ BMS transformations. Instead, these transformations are called ‘translations’ here simply because they correspond to an Abelian transformation on the volume variables.
- We have added a new appendix (now appendix A) with a table containing the list of some of the variables being used, as well as their role and relationship.
- We have added the labels news to the curves in all the plots.
- We have added the definition after the first appearance of ‘horocycle’ in the introduction.
- We have added a comment about the quantization ambiguities in LQC at the end of the second to last paragraph of the intro. We have also added footnote 10 in section 6.
- As pointed out by the referee, it is true that we start our analysis with the more general line element (2.2). This is however just because we are considering a phase space, and not only the particular solution (2.1). Equation (2.15) simply shows consistently that (2.1) is indeed the (only) solution within the phase space of (2.2).
- We have added a sentence below (3.10) to state which symmetry is associated with which charge.
- We have changed the signature in section 5 to mostly minus, so that it is now consistent with the rest of the paper. However, please note that the signature in section 5 is that of an auxiliary geometric space, and not that of spacetime as in the rest of the paper, so there is a priori no need for the two signatures to match, and we can choose them for convenience.
- We have added the limits in the caption and in the sentence below (5.17).
- We have replaced ‘consists in’ by ‘consists of’ in the relevant places.

---

## Round 1 · Referee Report · Anonymous · 2020-12-16

Strengths
1. The authors describe a new symmetry for black hole interiors, namely iso(2,1), that relates different black hole solutions, and they explore it in detail, from many complementary perspectives. The connection to the 3D BMS group is pointed out.
2. The authors use this symmetry to construct an effective loop quantum cosmology (LQC) framework for black hole interiors that preserves this symmetry. It is emphasized that this symmetry could be useful as a guiding principle for quantization, with this as a nice example.
3. The paper is clearly written.
Weaknesses
1. The main limitation of the paper is that it is based on the isometry between the interior of the Schwarzschild space-time and Kantowski-Sachs; this isometry holds in classical GR but it is not clear if it will also hold in quantum gravity, or not. In particular, if quantum gravity effects give an inner horizon (as suggested for example by Bardeen and many others after), then it will not be possible to identify the black hole interior with a Kantowski-Sachs space-time.
Report
This paper studies the interior of a Schwarzschild black hole, describing a new iso(2,1) algebra that relates black hole (interior) solutions with different masses. The authors argue that it may be important to preserve these symmetries in the quantum theory, and that this condition could be an important guiding principle. The authors follow this path to construct an effective loop quantum cosmology framework for the interior, and show that these effects resolve the black hole singularity and replace it by a transition to a white hole.
These are interesting results, which are nicely presented, that merit publication.
Requested changes
1. On pp. 3-4 and on p. 34, the authors suggest that the appearance of the 3D BMS group (rather than the 4D BMS group) may be due to the symmetry reduction to a minisuperspace model. It would be interesting to see if a similar symmetry is present for spherically symmetric space-times with a local (radial) degree of freedom. In particular, if there is a symmetry group in this context also, and it is a subgroup of the 4D BMS group, then this would support their claim. A short discussion on this point might be useful. Note that such an extension would also directly address the weakness mentioned above, namely that all of the results are based on an isometry of classical GR between the Schwarzschild interior and the Kantowski-Sachs space-time. And would it be possible to make contact with other work in LQC/LQG that studies the entire black hole space-time, interior and exterior?
2. In Eq. (A.17), it is explained how to choose the lapse so that the CVH algebra closes. Is such a choice generically possible? Or is it only because of the high degree of symmetry that this can be done? The authors should say more on this point. Also, it seems to me that this is a key step in the paper (this is the motivation underlying the choice of the lapse in Sec. 2.1 that is essential for the remainder of the paper), and it seems odd to place this explanation in an appendix. I encourage the authors to move this into the main text in Sec. 2.1.
Some minor points:
3. In Eq. (3.3), it would be helpful to give the explicit form for F.
4. In Eq. (3.9), the authors should add indices to each F.
5. In Eq. (3.10), the authors should explain that the +,-,0 subscripts are chosen for convenience later in Eq. (3.17), and explain how they are related to t,d,s transformations.
6. Similarly, the subscripts +,-,0 for Eqs. (3.14)-(3.15) are chosen for later convenience, but it would be helpful to relate each of these to polynomials of order 0, 1 and 2.
7. At the beginning of Sec. 4, the authors write "We have shown in the previous section that the Kantowski–Sachs mini-superspace for the black hole interior metric admits a conformal and translational invariance on top of the diffeomorphism symmetry of general relativity." It might be helpful to remind the readers here that the reason why it is clearly different from the residual diffeomorphism symmetry is that it relates physically different space-times (black holes with different masses) to each other, unlike diffeomorphisms.
8. In Eqs. (4.9), (4.12) and (4.20), the dot is not easily seen in the subscript 1/\dot{f}. It would be good to reformat this so it is easier to read.
9. Below Eq. (5.16), the authors write "This is due to the non-vanishing L_0 shift of the Hamiltonian density, which therefore really plays an essential mathematical role in the whole model despite having been introduced as a non-physical IR regulator." I think this is slightly misleading; the constant term in the Hamiltonian comes from the 3D Ricci scalar (integrated over the fiducial region, and up to some overall prefactors absorbed into the lapse). So this term has a clear physical interpretation. It is the overall scale of the prefactor L_0^2 that is non-physical, but this scaling is present in every term in the Hamiltonian, not just the constant term. This sentence should be rewritten so it is clear that the term coming from the 3D Ricci scalar is physical.
10. In Eq. (6.1), the \lambda_i are constants that do not depend on phase space variables. This should be made clear, and the authors should explain that as a result the effective theory is not based on the "\bar\mu" or "improved dynamics" scheme. It would be interesting if the authors could briefly comment on whether their prescription could also be used to implement \bar\mu dynamics in an effective loop quantum cosmology framework.

---

## Round 2 · Referee Report · Anonymous · 2021-1-18

Report

The authors have addressed all of my suggestions and I am happy to recommend publication.

Requested changes

I have one optional suggestion for the authors: it might be helpful to add a reference to Appendix B below Eq. (2.7) for a more detailed explanation on why this choice for the lapse leads to a closed CVH algebra.

---

## Round 2 · Referee Report · Anonymous · 2021-1-18

Report

The authors have satisfactorily addressed all my comments. Therefore, I am happy to recommend this interesting article for publication.

---

## Round 2 · Author Response

In this second version of the manuscript all the changed suggested by the referees have been implemented.

---

## Editorial Decision

published